# A Microwave Scattering Database of Oriented Ice and Snow Particles: Supporting Habit-Dependent Growth Models and Radar Applications (McRadar 1.0.0)

Leonie von Terzi[1], Davide Ori[2], and Stefan Kneifel[1]

[1]Institute of Meteorology, Ludwig-Maximilians University, Munich, Germany
[2]Institute of Geophysics and Meteorology, University of Cologne, Cologne, Germany

**Correspondence:** Leonie von Terzi (l.terzi@lmu.de)

**Abstract.**

The optical properties of atmospheric hydrometeors are a crucial component of any forward operator. These forward operators are essential data assimilation, and atmospheric model evaluations. Recent advances in microphysical modelling, such as Lagrangian super-particle models with habit prediction for ice particles, allow for the continuous evolution of particle properties in contrast to fixed hydrometer classes with fixed properties. This increasing complexity demands scattering databases capable of handling a wide range of particle properties.

The discrete dipole approximation (DDA) is one of the most accurate and widely used methods for computing the scattering properties of irregular ice particles. However, its high computational cost typically limits either the diversity of particle shapes or the range of environmental parameters (e.g., frequency, temperature) represented in existing databases, constraining their applicability to models with highly variable microphysics.

In this study, we present a new DDA-based database of optical properties at 5.6, 9.6, 35.6, and 94 GHz, specifically designed to accommodate the broad range of ice crystal morphologies predicted by habit-evolving schemes. The database contains 2,627 individual ice crystals, including dendrites, plates, and columns, as well as 450 aggregates with varying degrees of riming and crystal types. The data are organized in three levels: Level 0 provides raw scattering matrices at individual orientations for a full range of scattering angles; level 1a summarizes Mueller and amplitude matrix elements relevant for radar applications (at forward and backward scattering angles); and level 1b offers lookup tables of scattering properties that are relevant for polarimetric radars assuming azimuthally random orientations of the particles. These data allow for flexible treatment of the canting angle of the particles. The lookup tables are directly accessible via the McRadar simulator and can also be interfaced with other forward operators.

The new database allows for a more consistent and realistic treatment of evolving ice particle properties in atmospheric models, improving the interpretation of radar observations and model-observation integration.

## 1 Introduction

The ice phase in clouds plays a crucial role in numerical weather prediction and climate modelling, as approximately 75% of precipitation reaching the ground forms via ice-phase processes (Heymsfield et al., 2018). However, significant knowledge gaps remain regarding ice microphysical processes, leading to substantial uncertainties in their parametrizations within models (Morrison et al., 2020). Forward models operating in the microwave region are essential for the development of retrieval techniques, both for active and passive microwave observations, and for the assimilation of satellite data into models (Geer et al., 2018). A key component of such forward models is the availability of comprehensive databases of particle optical properties (Maahn et al., 2020).

Estimating the optical properties of ice particles is a challenging task. In the microwave region, accurately characterizing these properties requires detailed knowledge of the particles' microphysical characteristics, including size, mass, density, shape, internal structure, and composition. The complex structures of ice particles can be modelled using physical snow growth models that simulate depositional growth, aggregation, and riming, such as the aggregation model introduced by Leinonen and Moisseev (2015) and later extended by Karrer et al. (2020). Once the particle shape is determined, its optical properties must be computed using numerical methods. A widely used technique for calculating the polarimetric radar signatures of ice particles is the T-matrix method, which approximates ice particles as spheroids with a homogeneous mass distribution (Mishchenko et al., 1996). This allows for a flexible treatment of mass, size, aspect ratio and density, which is convenient when trying to represent various shapes of (ice) particles. However, real snowflakes typically exhibit highly inhomogeneous internal mass distributions that strongly influence their scattering characteristics (Sorensen, 2001). As a result, T-matrix methods often underestimate polarimetric signals for complex particles such as aggregates (e.g., Petty and Huang, 2010; Schrom and Kumjian, 2018; Kneifel et al., 2020).

The Discrete Dipole Approximation (DDA) (Draine and Flatau, 1994) has become one of the most widely used and sufficiently accurate methods for calculating the scattering properties of irregularly shaped particles. DDA discretizes the particle onto a grid of dipoles and can accommodate arbitrary shapes. It has been extensively validated (Yurkin and Hoekstra, 2007), but its high computational cost, scaling approximately with $O(n\log(n))$, where $n$ is the number of dipoles, remains a significant limitation. During the last decade, increased computational power has facilitated the development of DDA databases. Many of these databases (Liu, 2008; Kuo et al., 2016) assume randomly oriented ice particles, which limits their applicability to polarimetric radar observations. More recent efforts, such as Lu et al. (2016) and Brath et al. (2020), have provided scattering properties across a range of orientations. For example, the Lu et al. (2016) database includes over 1,000 ice particles, ranging from single crystals to aggregates and graupel, at X-, Ku-, Ka-, and W-band frequencies. Brath et al. (2020) focused on a smaller number of particles but extended the frequency range, supporting applications up to the sub-mm region.

In-situ observations (Bailey and Hallett, 2009) and laboratory studies (Pruppacher and Klett, 1997) revealed a wide variety of shapes that ice particles can grow into by initial depositional growth. The shape is mainly determined by the temperature and super-saturation of the environment in which the particles grow. The shape of ice crystals influences the particles' fall velocity, the growth rate by deposition, as well as subsequent collisional processes such as riming and aggregation. Those changes in size, shape, density, and orientation also impact the particles' interaction with radiation, which is the reason for distinct radar signatures of ice particles in different regions of the cloud. In most bulk microphysical schemes this variety of microphysical properties is typically represented by only a few fixed ice categories (e.g.,

ice, snow, graupel). The Predicted Particle Properties (P3) scheme (Morrison et al., 2025) overcomes this limitation by employing a single
ice category in which the physical properties are predicted directly from environmental conditions. While in the P3 scheme, the particle
shape is still fixed, the morphological evolution of ice crystals is predicted by habit prediction schemes (Welss et al., 2024; Jensen et al.,
2017; Jensen and Harrington, 2015; Harrington et al., 2013) (see Section 5.1). Due to high computation costs, habit prediction schemes have
been mainly applied in experimental models such as Lagrangian super-particle models (Shima et al., 2009), however, Jensen et al. (2017)
have introduced the bulk microphysics scheme ISHMAEL which predicts the evolution of ice properties, including the aspect ratio of the
particles. An example of a recent Lagrangian super-particle model with ice habit prediction included is the McSnow model (Brdar and Seifert,
2018; Welss et al., 2024). A major advantage of Lagrangian models is the possibility to trace the particle history and morphological change
caused by microphysical processes. Such models are considered to be important tools to derive improved microphysical parametrizations
for bulk schemes (Morrison et al., 2020). In order to evaluate such new modelling approaches, for example, with radar observations,
scattering databases are needed that consistently cover the wide range of ice particle properties predicted by the model. A database with
a wide variety (mass, aspect ratio, size) of spheroids with homogeneous ice-air mixture would be consistent with the assumptions in habit
prediction schemes, but the non-homogeneous mass distribution in real ice particles impacts their scattering properties (Sorensen, 2001;
Kneifel et al., 2020). The true in-cloud geometries of ice particles is vast and remains unknown, thus creating large uncertainties when
generating "representative" snow particles for forward simulations.

These uncertainties were investigated by Schrom and Kumjian (2019). They developed a probabilistic forward operator that maps
microphysical model output to radar observations using both DDA-derived scattering properties and equivalent spheroidal approximations.
Their method involved statistically fitting equivalent spheroids to match the backscatter cross-section at horizontal polarization ($\sigma_h$) and the
differential reflectivity (ZDR) of complex crystals. This mapping enabled the quantification of uncertainties arising from dendritic branching;
for example, a standard deviation of $\pm 0.5$ dB in ZDR at X-band was reported.

Even greater uncertainties are expected for snow aggregates, which consist of multiple ice crystals with varied shapes and orientations. Ori
et al. (2021) demonstrated the potential bias introduced when using a single representative aggregate for a given size or mass by comparing
single-particle DDA results to ensemble averages.

Both Schrom and Kumjian (2019) and Ori et al. (2021) emphasize the need for a scattering database that supports forward simulation of
complex microphysical model output, while minimizing errors associated with spheroidal approximations or limited particle shape diversity.
In this study, we present a new DDA-based scattering database that addresses this need. The database includes the scattering properties of
3,077 ice particles at C- (5.6 GHz), X- (9.6 GHz), Ka- (35.6 GHz), and W-band (94 GHz) frequencies. Scattering properties are provided for
elevation angles from -90° to 90° in 5° steps and azimuth angles from 0° to 360° in 22.5° steps. The database includes 2,627 ice crystals,
encompassing dendrites, plates, and needles, as well as 450 aggregates with varying crystal types and degrees of riming.

The forward and backward scattering properties at all elevations and azimuth directions of all particles are freely available[1]. Additionally,
we provide averages of azimuthally random orientations for each particle to enable flexible canting-angle assumptions in forward operators[2].
In this study, we also introduce a forward operator (Section 4.1) that maps microphysical particle properties into radar observables. Unlike
Schrom and Kumjian (2019), our approach uses ensemble-averaged scattering properties, derived by averaging over a number of nearest
neighbours in microphysical property space. Finally, we present two application examples that illustrate the strengths of this new database.

---

[1]see https://doi.org/10.57970/cgtek-fen7
[2]see https://doi.org/10.5281/zenodo.16792943

## 2 Ice Particle Properties

The habit of ice crystals varies depending on the ambient temperature and supersaturation (Libbrecht, 2017). Recent advancements in bin and Lagrangian particle microphysical models, including those with habit prediction capabilities, have created a demand for scattering databases that allow consistent forward simulation of model output. It is well known that the internal structure of ice particles is essential for scattering in the microwave regime (e.g. Kneifel et al., 2020; Ori et al., 2021; Brath et al., 2020). Realistically shaped ice particles can be simulated using so called aggregation models (Leinonen and Moisseev, 2015; Leinonen and Szyrmer, 2015; Westbrook et al., 2004a, b). The aggregation model introduced in Leinonen and Moisseev (2015) and Leinonen and Szyrmer (2015) constructs three-dimensional representations of aggregate snowflakes by combining crystals of predefined shapes such as hexagonal columns, plates and dendrites. The aggregation model is conceptional similar to work provided by Westbrook et al. (2004a, b). The same aggregation model has been used in various studies, for example as the input into complex scattering calculations (Leinonen et al., 2018; Ori et al., 2021; Maherndl et al., 2023), to investigate the geometry of rimed ice particles (Seifert et al., 2019), to evaluate parametrizations of ice and snow in models (Karrer et al., 2020; Ori et al., 2020), to create retrieval methods based on in-situ and radar observations (Maherndl et al., 2025; Leinonen et al., 2021) and to generate 3D-printed snowflakes for laboratory studies (Köbschall et al., 2023) . In this study, we have used the aggregation model to develop a comprehensive scattering database containing 3,077 ice particles, including 2,627 ice crystals and 450 aggregates. While the aggregate shapes were taken from previous studies (Ori et al., 2021; Karrer et al., 2020), the ice crystal shapes were specifically created for this database.

### 2.1 Ice Crystals

In the dendritic growth layer (corresponding to temperatures between $-20^{\circ}$C and $-10^{\circ}$C), supersaturation levels significantly influence the degree of branching in plate-like crystals, resulting in a wide variety of morphologies (Bailey and Hallett, 2004, 2009; Lohmann et al., 2016; Takahashi, 2014; Libbrecht, 2017). Takahashi (2014) categorized these forms as sector, broad-branch, stellar, dendritic, and fern-like structures. This high degree of complexity of the internal structure of dendritic ice crystals cannot be explicitly described in habit prediction schemes such as the one implemented in Welss et al. (2024). The habit prediction schemes therefore describe ice crystals as porous spheroids, whose aspect ratio and density is varied by the habit prediction. A detailed description is given in Section 5.1. While the description of ice crystals as spheroids gives reasonably well results when comparing the aspect ratio, mass and maximum dimension ($D_{max}$) to laboratory studies (Welss et al., 2024), it is well known that the internal structure of ice particles is essential for scattering in the microwave regime (e.g. Kneifel et al., 2020; Ori et al., 2021; Brath et al., 2020). In order to forward simulate the output of models using habit prediction schemes, a large variety of ice crystal shapes has to be used, in order to match the predicted aspect ratios, densities, masses, and $D_{max}$.

To capture the morphological diversity of dendritic ice crystals, we therefore employed the Reiter-algorithm (Reiter, 2005) as implemented in the aggregation model. The Reiter algorithm is a two-dimensional cellular automaton defined on a hexagonal lattice. Each grid cell represents vapour, boundary, or ice and contains a scalar supersaturation value. The evolution of the crystal proceeds in discrete time steps consisting of a diffusion step and an attachment (growth) step. In the diffusion step, supersaturation is redistributed between neighbouring cells using a weighted average controlled by the parameter $\alpha$, which specifies the relative strength of diffusion. Higher values of $\alpha$ lead to faster smoothing of the supersaturation field, resulting in more open, dendritic structures, while smaller values promote more compact shapes.

Boundary cells, those adjacent to ice, can accumulate mass from the surrounding supersaturation field. The rate of this accumulation is governed by $\beta$, which sets how quickly vapour condenses onto the growing crystal edges. Once the accumulated mass of a boundary

cell exceeds a threshold determined by $\gamma$, the cell is converted into ice. The parameter $\gamma$ therefore controls the overall growth rate and the

sharpness of branching features: smaller $\gamma$ values allow rapid conversion to ice and more finely branched structures, whereas larger $\gamma$ values promote slower, smoother growth. When $\beta = 0$ and $\gamma > 0$, plates are formed, and regardless of $\beta$, at high $\gamma$, plates, or broad branched stellar forms are produced. For levels of low $\gamma$ and $\beta$, dendritic growth happens, while for larger values of $\beta$, stellar forms appear. Figure 3 in Reiter (2005) shows an overview of the effect of $\gamma$ and $\beta$ on the form of the crystal.

Iterating the diffusion and attachment rules yields crystal patterns that exhibit hexagonal symmetry, dendritic branching, and sector

structures consistent with natural snow crystals. Despite its conceptual simplicity, the Reiter algorithm captures the principal morphological characteristics of atmospheric ice particles.

We used the Reiter algorithm to generate a large variety of dendritic shapes. Specifically, we set $\alpha = 1$ and varied $\beta$ between 0.2 and 0.7 and $\gamma$ between 0.0001 and 0.006. $D_{\max}$ ranged from 10 $\mu$m to 7 mm. Figure 1 presents a subset of the resulting ice crystal shapes.

The variety in aspect ratios generated by the Reiter algorithm alone is not enough to cover the entire aspect ratio space predicted by the

habit prediction. In extreme cases, a particle can change growth regime from e.g., the columnar growth regime assumed at temperatures colder than $-22°$ C into the plate-like growth regime at temperatures between $-20$ and $-10°$ C. Bailey and Hallett (2004) have shown that when such a regime change happens, the particle is likely to decrease its aspect ratio towards unity, effectively generating a "thick" column. To capture these particles, we adapted the aggregation model from Leinonen and Moisseev (2015) to allow the generation of ice crystals with adjustable aspect ratios at fixed $D_{\max}$. This was done by effectively stacking dendritic particles of the same shape on top of each other,

increasing the aspect ratio towards unity. In total, we generated 1,850 dendritic ice crystals.

At smaller sizes or under low supersaturation conditions, solid plates are more likely to form. To represent this regime, we generated 625 plate crystals with varying sizes, masses, and aspect ratios with the aggregation model and our adaptation to allow more flexible aspect ratios. In the temperature range between $-10°$C and $-5°$C, ice crystals tend to grow into columnar shapes (Bailey and Hallett, 2009). Accordingly, we used the aggregation model to generate 125 columnar crystals, again varying size, mass, and aspect ratio. A description of the plates and

columns is given in Figure 1 of Leinonen and Moisseev (2015).

The resulting mass–size and aspect ratio–size relationships for all 2,600 ice crystals are illustrated in Figure 2.

For the preceding scattering calculations, the particles are projected onto a regular Cartesian grid. In order to capture even small details of the particle shape, we chose a grid resolution dependent on $D_{\max}$ such that $grid_{res} = D_{\max} \cdot 2.5 \cdot 10^{-2}$.

## 2.2   Aggregates

Similarly to ice crystals, the microphysical properties of aggregates vary largely depending on the ice crystals which generated the aggregate and stochastic parameters determined during the collision process, such as the angle between the colliding partners, or the contact point (e.g. Locatelli and Hobbs, 1974; Mitchell et al., 1996; Westbrook et al., 2004b). Therefore, we have randomly selected 450 aggregates of plates, dendrites, mixtures of dendrites and columns (mix2) with a riming equivalent liquid water content (see Leinonen and Moisseev (2015)) between 0 and 1kg m$^{-2}$ from the snow library of snowScatt (Ori et al., 2021). The resulting database comprises 175 rimed particles, 106

mixtures of columns and dendrites, 105 dendrite aggregates, and 63 plate aggregates. A subset of their shapes in presented in Figure 3, the microphysical properties of all aggregates are shown in Fig. 4.

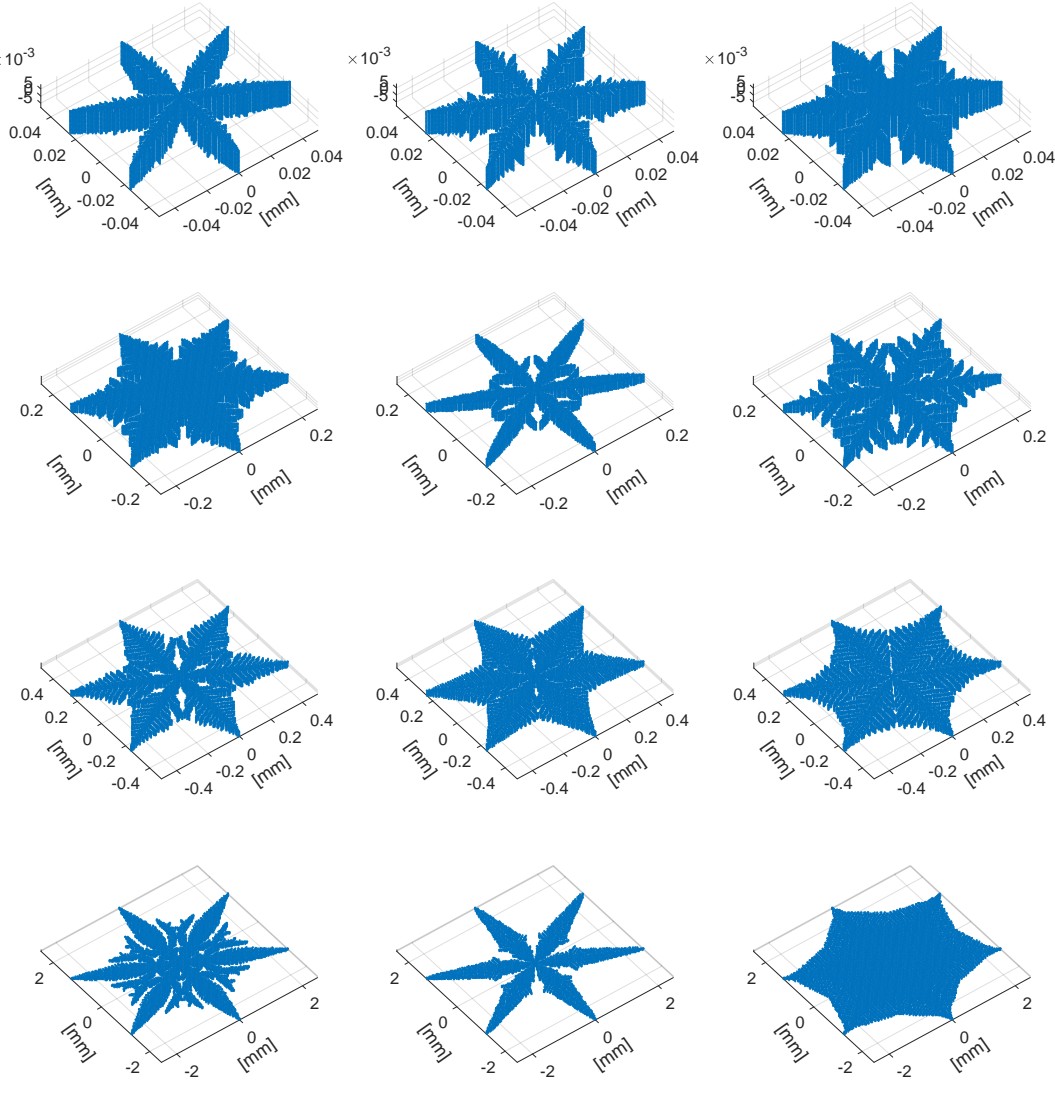

**Figure 1.** Examples of dendritic ice crystal shapes from the scattering database. The first row shows dendrites with a $D_{max}$ of 100 $\mu$m, the second row with 500 $\mu$m, the third row with 1 mm and the last row with 5 mm.

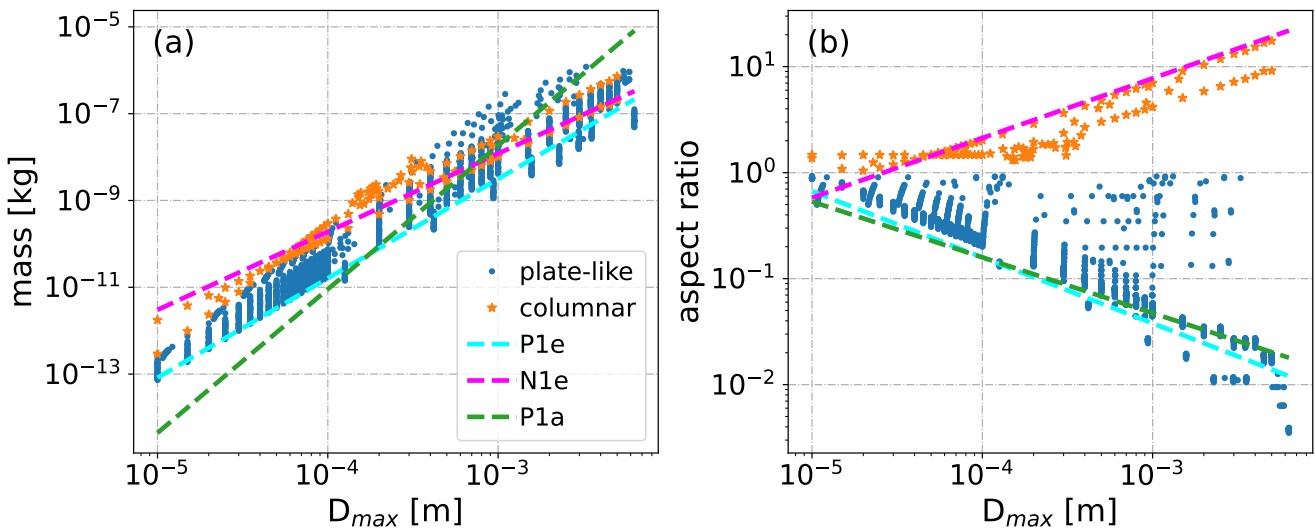

**Figure 2.** Ice particle mass (a) and aspect ratio (b) as a function of the maximum dimension ($D_{max}$). Orange asterisks denote columnar, blue dots denote plate-like particles. For comparison, the relationships of the Pruppacher and Klett (1997), plate (P1a), dendrite (P1e) and needle (N1e) are shown.

## 3 Radar scattering properties

### 3.1 Discrete Dipole Approximation

After determining the particle shapes, their scattering properties must be computed. The discrete dipole approximation (DDA) is a widely used method for calculating the scattering of arbitrarily shaped particles by solving the volume integral equation form of Maxwell's equations for a discretized representation of each particle. Originally developed by Purcell and Pennypacker (1973), the method replaces the particle with an array of polarizable dipoles. The interactions among these dipoles and with the incident electromagnetic field yield a system of linear equations, the solutions of which give the dipole polarizations and thus allows to compute the scattered electromagnetic field.

Since its initial development, increases in computational power and the availability of efficient public implementations such as ADDA (Yurkin and Hoekstra, 2011) and DDSCAT (Draine and Flatau, 2013) have made the DDA a popular choice for modelling particles of arbitrary shape and composition. The accuracy of the DDA is considered to be limited by two sources of error, namely the *shape error* and the *discretization error* which are both dependent on the dipole size (Yurkin et al., 2006). These two sources of error refer to the accuracy at which a certain shape can be approximated by a set of cubic voxels and how well those same cubic volumes can represent the spatially varying electromagnetic field. A commonly used criterion for the latter is to guarantee that the parameter $|m|kd < 0.05$, where $m$ is the refractive index, $k$ the wavenumber, and $d$ is the dipole spacing (Tyynelä et al., 2009; Leinonen and Moisseev, 2015; Zubko et al., 2010; Yurkin and Hoekstra, 2007). At W-Band (94 GHz, the largest frequency in this database) this implies a dipole spacing below 14 $\mu$m; our largest dipole spacing of 10 $\mu$m satisfies this requirement.

The *shape error* is particularly critical for strongly asymmetric shapes, such as thin plates, whose thickness may span only a single layer of dipoles or heavily-branched dendrites with even finer internal structures. For this reason, we adopt a variable dipole spacing that depends on

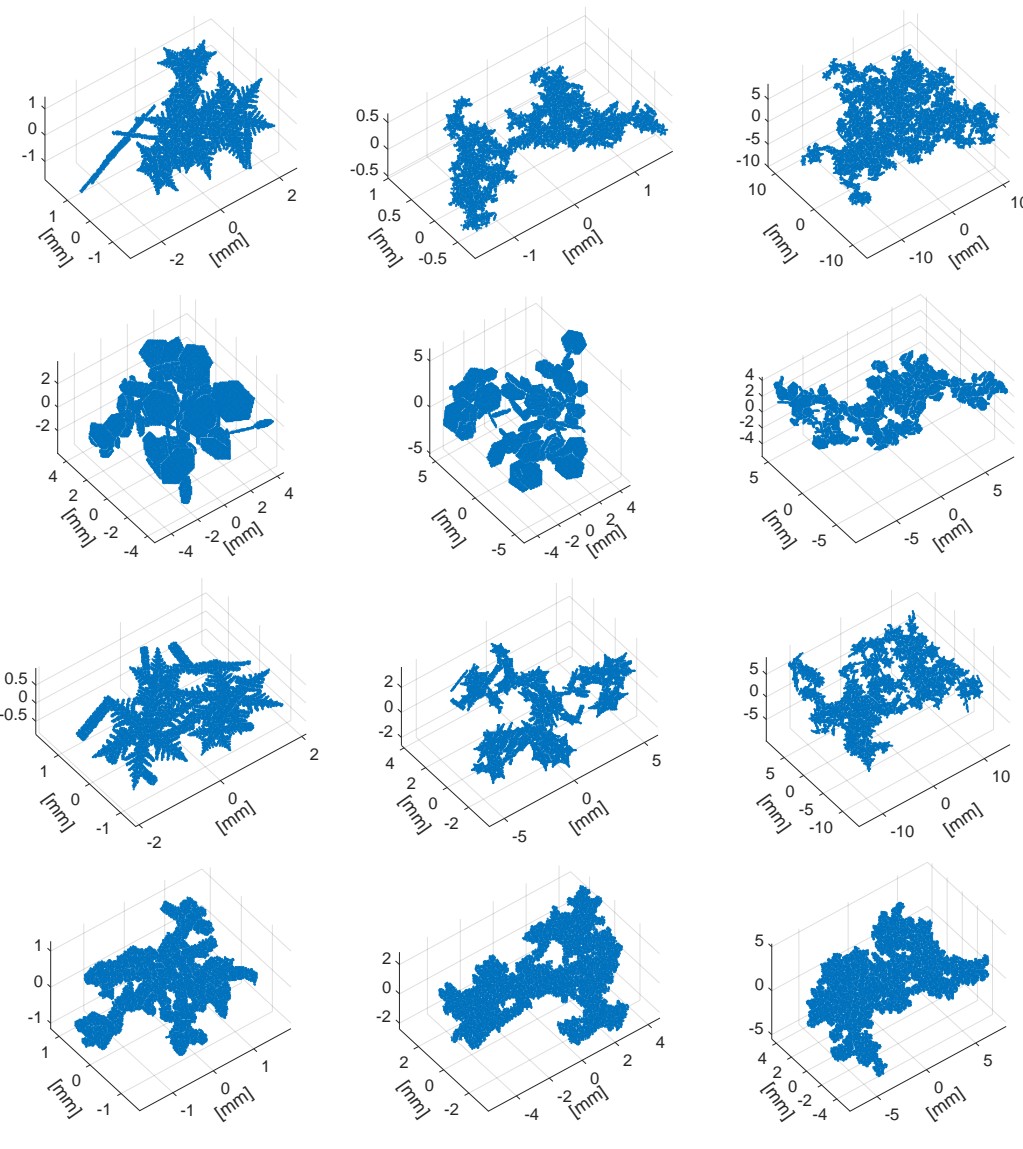

**Figure 3.** Examples of aggregates available in the scattering database. Shown are dendrite aggregates (top row), plate aggregates (second row), aggregates consisting of a mixtures of dendrites and columns (mix2) (third row) and rimed mix2 aggregates (fourth row).

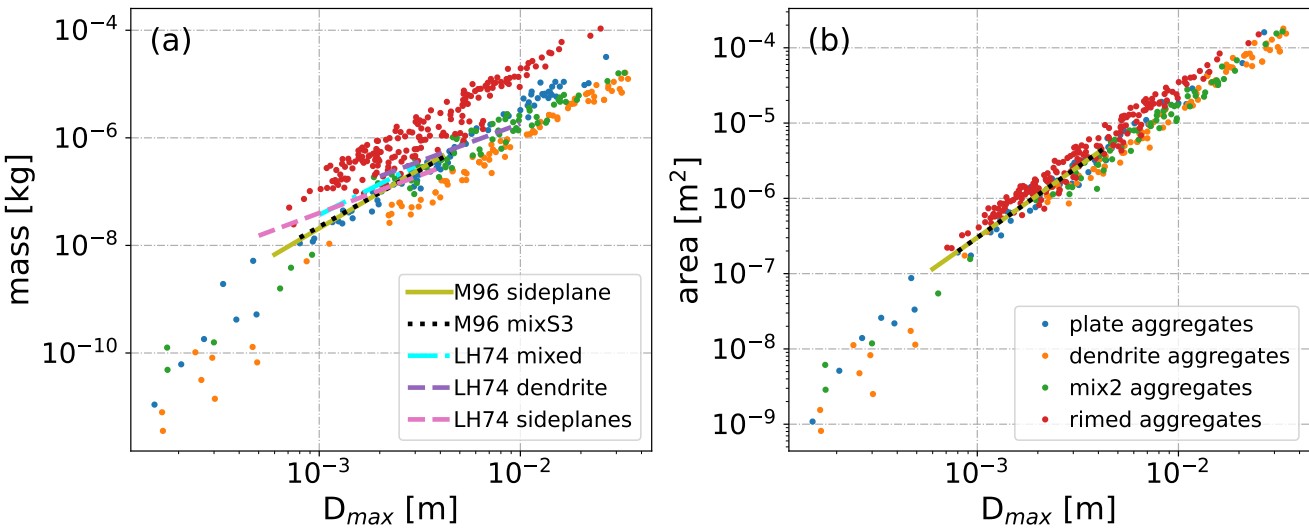

**Figure 4.** Aggregate mass (a) and cross-sectional area (b) as function of $D_{max}$. For comparison, the relationships of Mitchell (1996) for aggregates of side planes (M96 sideplane), aggregates of side planes, columns and bullets (M96 mixS3) as well as the relationships of Locatelli and Hobbs (1974) for aggregates of unrimed radiating assemblages of dendrites (LH74 dendrite), aggregates of unrimed radiating assemblages of plates, side planes, bullets and columns (LH74 mixed) and aggregates of unrimed side planes (LH74 sideplanes) are shown. Note: area-size relationships are only available for the Mitchell et al. (1996) aggregates.

particle size. Some studies (Yurkin et al., 2006; Ori and Kneifel, 2018), adopted specific investigation techniques to analyse these two error sources by varying the dipole resolution. Since this resolution is size-dependent in our database a comprehensive application of this approach to all shapes would be particularly effort-taking. Instead, we limited our analysis on two random dendritic particles from our database and reduced the spacing by factors of two and four. The resulting changes in the computed scattering properties were only 0.12–0.2%, depending on polarization and wavelength, indicating that the discretization is sufficient.

We would like to emphasize that uncertainties arising from DDA discretization are likely negligible compared with the much larger uncertainties associated with selecting appropriate particle shapes, as the true in-cloud geometries are unknown. We attempt to mitigate these uncertainties by simulating a wide variety of particles, but the error inherent in choosing any particular shape remains unknown and cannot currently be constrained by in-situ observations.

The full ADDA setup used for the simulations can be inferred from the shared analysis codebase and has been derived from the
experience gained in our research group regarding the calculation of the scattering properties of complex shaped frozen and partially melted hydrometeors (Ori et al., 2014; Ori and Kneifel, 2018; Ori et al., 2021). In particular Ori and Kneifel (2018) provides an overview of the performance of various combinations of ADDA configurations for such scattering targets. In particular we used Lattice Dispersion Relation (LDR) and Filtered Coupled Dipoles (FCD) respectively for polarization formulation and interaction term.

## 3.2 Reference frames

We first define two fundamental reference frames, namely the radar reference frame (RRF), and the particle reference frame (PRF). The RRF (often called laboratory reference frame) can be considered as the most general one, it is fixed with a hypothetical ground-based radar, the z-axis goes along the vertical with respect to the radar location, while the orientation of the horizontal xy-axes is arbitrary. The radar reference frame defines the coordinates of the propagation direction of the radar beam and its polarization planes (Fig. 5a). The propagation direction of the radar beam is defined by the radar elevation angle $el$ and azimuth $az$. The vertical polarization plane spans the beam propagation direction and the z-axis, while the horizontal polarization plane is orthogonal to the vertical and also contains the beam propagation vector.

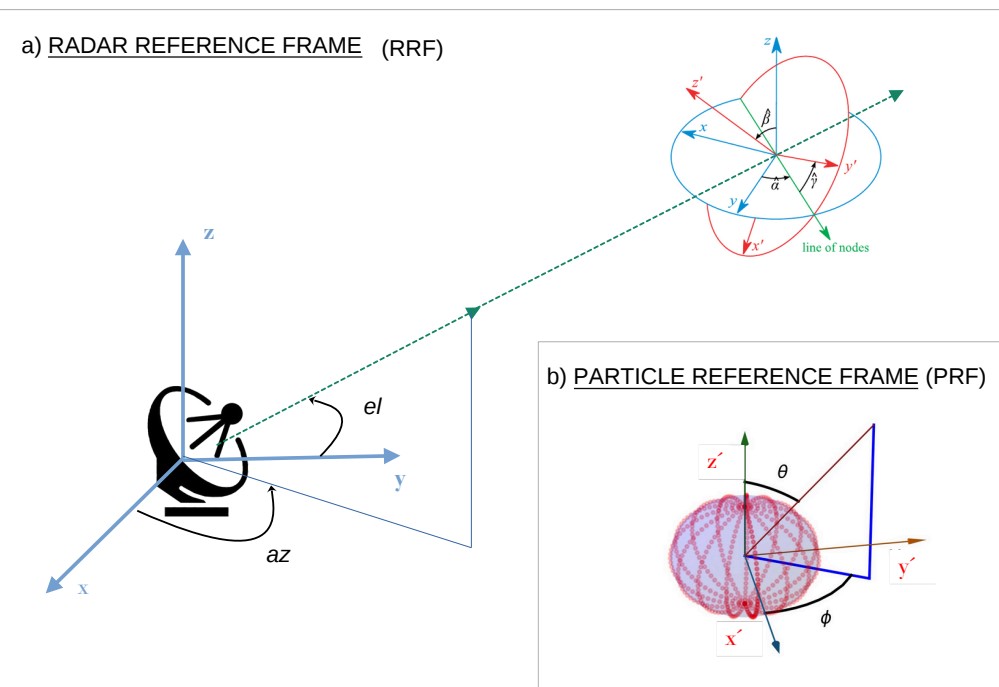

**Figure 5.** Panel a, Schematic representation of the Radar Reference Frame (RRF in blue) and the rotation of the Particle Reference Frame (PRF in red) with respect to the RRF employing the three Euler angles ($\hat{\alpha}$, $\hat{\beta}$ and $\hat{\gamma}$). Panel b, Detail of the unrotated PRF and the polar coordinates ($\theta$,$\phi$) that define any unit vector in the PRF. The red dots represent the set of sampling directions used in section 3.5 to describe the particle scattering in all possible orientations. The figure of the PRF and relative rotations with respect to the RRF is adapted from the ADDA User Manual.

The PRF is fixed with the snow particle, and is assumed to correspond to the particle axes of inertia with the shortest inertia axis aligned with the z-axis of the PRF. The transformation from PRF to RRF is performed by means of 3 Euler rotations (angles $\hat{\alpha}$, $\hat{\beta}$, and $\hat{\gamma}$) that follow the commonly used zyz-convention (Fig. 5b). With respect to the PRF, the propagation direction of any vector is defined by the polar coordinates $\theta$ and $\phi$.

The polarimetric scattering properties of a complex-shaped snowflake in the RRF would in general depend on all 5 angles ($el$, $az$, $\hat{\alpha}$, $\hat{\beta}$, $\hat{\gamma}$). Note that a rotation of the angles $az$ of the RRF and $\hat{\alpha}$ of the PRF act on axes that are parallel, which implies that only the difference $az - \hat{\alpha}$ is relevant for the definition of the overall geometry. By assuming $az = 0$ the dependence on the angle $az$ can be dropped without loss of information.

## 3.3 Single-scattering particle properties

At a given particle orientation, the incident electromagnetic field $E^i$ can be related to the scattered (far-) field $E^s$ via the amplitude scattering matrix (Mishchenko et al., 2002):

$$
\begin{bmatrix} E_\parallel^s \\ E_\perp^s \end{bmatrix} = \frac{\exp(ikr)}{r} \begin{bmatrix} S_1 & S_4 \\ S_3 & S_2 \end{bmatrix} \begin{bmatrix} E_\parallel^i \\ E_\perp^i \end{bmatrix}
\tag{1}
$$

the subscripts $\parallel$ and $\perp$ indicate parallel and perpendicular orientations with respect to the scattering plane, $i = \sqrt{-1}$, $k = 2\pi/\lambda$ is the wavenumber, $r$ is the distance between the scatterer and the observation point, and $S_{1,\ldots,4}$ are the complex elements of the amplitude matrix. Please note that the ADDA code outputs a slightly different definition of the amplitude matrix that includes a $-ik$ factor in the denominator. The amplitude matrix can be converted to the Mueller matrix $Z$, which relates the Stokes vector of the incoming and scattered fields (Mishchenko et al., 2002):

$$
\begin{bmatrix} I_{sca} \\ Q_{sca} \\ U_{sca} \\ V_{sca} \end{bmatrix} = \frac{1}{r^2} \begin{bmatrix} Z_{11} & Z_{12} & Z_{13} & Z_{14} \\ Z_{21} & Z_{22} & Z_{23} & Z_{24} \\ Z_{31} & Z_{32} & Z_{33} & Z_{34} \\ Z_{41} & Z_{42} & Z_{43} & Z_{44} \end{bmatrix} \begin{bmatrix} I_{in} \\ Q_{in} \\ U_{in} \\ V_{in} \end{bmatrix}
\tag{2}
$$

where $Z_{11,\ldots,44}$ are the real elements of the Mueller matrix. The Stokes vector consists of the elements I, Q, U and V. I describes the total intensity of the incoming/scattered field, Q and U which describe the linear polarization, in case of Q the preference of horizontal versus vertical polarization, and in case of U the preference of $\pm45°$ polarization. V describes the circular polarization. In contrast to the Amplitude matrix, the Mueller matrix contains only information about the intensity of the scattered wave, not the absolute phase. The Amplitude matrix can be converted into the Mueller matrix following well established formulas. Both matrices were calculated with the ADDA version 1.4.0 and are available in the scattering database at C- (5.6 GHz), X- (9.6 GHz), Ka- (35.6 GHz) and W- (94.0 GHz) band as well as for $el$ between 0° and 180° in 5° steps and $az$ between 0° to 337.5° in 22.5° steps for every scattering direction. Such a coarse azimuthal sampling is sufficient because the scattering properties are later averaged over azimuth (see Section 3.4). An azimuthal resolution as coarse as 45° yields identical azimuthally averaged scattering properties. The scattering properties were calculated at a temperature of 270 K, using the refractive index model described in Mätzler (2006), Chapter 5 for ice.

The full set of formulas to calculate the $Z_{11,\ldots,44}$ elements from the amplitude matrix can be found in Mishchenko et al. (2002) and in the code we shared. For convenience we report here the equations for the four elements that are relevant for the radar quantities we calculate in McRadar

$$
Z_{11} = \frac{1}{2}(|S_1|^2 + |S_4|^2 + |S_3|^2 + |S_2|^2)
\tag{3}
$$

$$
Z_{12} = \frac{1}{2}(|S_1|^2 - |S_4|^2 + |S_3|^2 - |S_2|^2)
\tag{4}
$$

$$Z_{21} = -\text{Re}\left(S_1 \cdot S_4^* + S_2 \cdot S_3^*\right) \tag{5}$$

$$Z_{22} = -\text{Im}\left(S_1 \cdot S_4^* - S_2 \cdot S_3^*\right) \tag{6}$$

where the asterix apex denotes the complex conjugate while $\text{Im}(var), \text{Re}(var)$ the imaginary and real parts of the variable $var$ respectively.

240      After deriving the amplitude and mueller scattering matrices using the Mishchenko et al. (2002) definition, typical radar parameters can be calculated as follows (Mishchenko et al., 2000):

- Radar reflectivity factor at horizontal polarization

$$Ze_{\text{hh}} = \frac{2\lambda^4}{\pi^4 |K|^2} \left(Z_{11}|_{180°} + Z_{22}|_{180°} + Z_{12}|_{180°} + Z_{21}|_{180°}\right) \tag{7}$$

- Radar reflectivity factor at vertical polarization

245
$$Ze_{\text{vv}} = \frac{2\lambda^4}{\pi^4 |K|^2} \left(Z_{11}|_{180°} + Z_{22}|_{180°} - Z_{12}|_{180°} - Z_{21}|_{180°}\right) \tag{8}$$

- Extinction cross-section at horizontal and vertical polarization:

$$c_{\text{ext,hh}} = \frac{4\pi\text{Im}\left(S_2|_{0°}\right)}{k}; \quad c_{\text{ext,vv}} = \frac{4\pi\text{Im}\left(S_1|_{0°}\right)}{k} \tag{9}$$

- Specific differential phase shift:

$$\text{KDP} = \frac{180\lambda\left(\text{Re}(S_2|_{0°}) - \text{Re}(S_1|_{0°})\right)}{\pi} \tag{10}$$

250 where $\lambda$ is the wavelength, $|K|^2$ the dielectric factor and the subscript $0°, 180°$ refers to the scattering direction, with $0°$ being forward and $180°$ backward.

## 3.4    Orientation averaging

Radar polarimetric quantities are sensitive to the shape and orientation of the particles. The steady-state falling orientation of a complex-shaped particle, such as a snowflake, is determined by the complex equilibrium of gravitational and aerodynamic forces. The problem of 255 finding the fall attitude of a snowflake is further complicated by the fact that such a stationary orientation might not even exist (i.e., snowflakes can continuously tumble and spin (McCorquodale and Westbrook, 2021)), furthermore, hydrometeors typically fall in an environment that is itself non-stationary and turbulent.

     The attitude of a snowflake during its fall can be described by the probability distribution of the three Euler angles that define its orientation with respect to the default fall configuration. For this study, the default orientation is defined as aligning the longest axis of inertia horizontally 260 (i.e. this corresponds to the Euler angle $\hat{\beta} = 0$). The distributions of Euler angles defining the particle orientations are commonly defined by $f_{\hat{\alpha}}(\hat{\alpha})$, $f_{\hat{\beta}}(\hat{\beta})$, $f_{\hat{\gamma}}(\hat{\gamma})$, but please beware that it is not always possible to factorize the probability distribution of the particle orientation on the product of the individual distribution of the three Euler angles. The average value of a quantity $\mathbf{S}(\hat{\alpha}, \hat{\beta}, \hat{\gamma})$ that depends on the orientation of the particle can be calculated as Brath et al. (2020)

$$\langle \mathbf{S} \rangle_{\hat{\alpha},\hat{\beta},\hat{\gamma}} = \int\limits_0^{2\pi} \int\limits_0^{\pi} \int\limits_0^{2\pi} \mathbf{S}(\hat{\alpha}, \hat{\beta}, \hat{\gamma}) f_{\hat{\alpha}}(\hat{\alpha}) f_{\hat{\beta}}(\hat{\beta}) \sin(\hat{\beta}) f_{\hat{\gamma}}(\hat{\gamma}) \, \mathrm{d}\hat{\alpha} \, \mathrm{d}\hat{\beta} \, \mathrm{d}\hat{\gamma} \tag{11}$$

A radar observes a vast population of hydrometeors whose orientation in space can be defined by angular distributions. They also sense those particles over a certain amount of time, during which particles change their orientation through, i.e. tumbling and spinning movements. For this reason we are not interested in the scattering properties of a single oriented particle. Rather, we are interested in orientation-averaged scattering properties of the particles population. This can be represented as an ensemble of multiple particles oriented differently but otherwise identical. It is widely accepted that the $\hat{\alpha}$ and $\hat{\gamma}$ angles follow a uniform distribution (Mishchenko and Yurkin, 2017). Averaging over the distributions of $\hat{\alpha}$ and $\hat{\gamma}$ leads to the representation of polarimetric scattering properties that follow the so-called *azimuthally random orientation* (ARO) assumption (Brath et al., 2020). Under these conditions, the polarimetric scattering properties of snowflakes depend on only two angles, namely the radar elevation angle $el$ and the particle canting angle $\hat{\beta}$.

$$S_{\mathrm{ARO}}(el,\hat{\beta}) = \int\limits_0^{2\pi} \int\limits_0^{2\pi} \frac{S^{\mathrm{RRF}}(el,\hat{\alpha},\hat{\beta},\hat{\gamma})}{4\pi^2} \, \mathrm{d}\hat{\alpha} \, \mathrm{d}\hat{\gamma} \tag{12}$$

### 3.5 Sampling discrete scattering orientations

In the previous section, we explained how it is possible to leverage the assumption that hydrometeors have no preferential orientation in the azimuth to reduce the dimensionality of the polarimetric scattering data structure. However, to compute Eq. 12 one would still need to perform a sufficient number of scattering calculations on individual sets of elevation angles and particle orientations $(el,\hat{\alpha},\hat{\beta},\hat{\gamma})$. This is a considerable computational effort, thus, following previous studies (Brath et al., 2020; Lu et al., 2016) we reduce the dimensionality of the problem by realizing that any scattering problem in the RRF (identified by the five angles $el$, $az$, $\hat{\alpha}$, $\hat{\beta}$, $\hat{\gamma}$) can be represented by a corresponding scattering in the PRF, provided that one can operate a proper transformation between the two reference frames. In the PRF any scattering problem is identified simply by the two polar coordinates $(\theta_0,\phi_0)$ that define the propagation direction of the incoming electromagnetic wave in the PRF. Mathematically $(el,az,\hat{\alpha},\hat{\beta},\hat{\gamma}) \Rightarrow (\theta_0,\phi_0)$.

The transformation matrices $\rho$ and $\rho^{-1}$ that allow one to convert the amplitude matrices from the PRF to the RRF are provided by Mishchenko (2000) and depend on the polar coordinates of the direction of the incoming and scattered beams as well as the three Euler angles that define the orientation of the scatterer. Here, we drop the dependency on the scattered directions for the sake of simplicity, since, for radar applications, the only two relevant scattering directions are the forward and backward ones. Mathematically, one can write

$$\mathbf{S}^{\mathrm{RRF}}(el,az,\hat{\alpha},\hat{\beta},\hat{\gamma}) = \rho^T(el,az,\hat{\alpha},\hat{\beta},\hat{\gamma}) \mathbf{S}^{\mathrm{PRF}}(\theta_0,\phi_0) \rho(el,az,\hat{\alpha},\hat{\beta},\hat{\gamma}) \tag{13}$$

Thus, we perform individual scattering calculations in the PRF with a regular $(\theta_0,\phi_0)$ grid. The polar angle $\theta_0$ ranges from -90° to 90° with increments of 5° while the azimuth angle $\phi_0$ ranges from 0 to 360° with increments of 22.5°. The set of directions used is represented as red dots in Fig. 5b. This approach is less efficient than other strategies to sample points over a unitary sphere (such as the icosahedral sampling strategy of Brath et al. (2020)), because it would allow for an increased density of sampling points close to the poles. However, it is fundamentally simple and has the advantage of integrating perfectly over azimuths in the case of non-tumbling ($\hat{\beta}=0$) particles.

The azimuth averaging of Eq. 12 is then performed numerically by sampling the database of individual scattering directions using a nearest-neighbour approach, hence $(el,az,\hat{\alpha},\hat{\beta},\hat{\gamma}) \Rightarrow (\theta_0^{NN},\phi_0^{NN})$. The transformation of Eq. 13 is then applied to the sampled matrix before executing the quadrature rule which is equivalent to perform the following integral

$$\mathbf{S}_{\mathrm{ARO}}(el,\hat{\beta}) = \int\limits_0^{2\pi} \int\limits_0^{2\pi} \frac{\rho^T(el,az,\hat{\alpha},\hat{\beta},\hat{\gamma}) \mathbf{S}^{\mathrm{PRF}}(\theta_0^{\mathrm{NN}},\phi_0^{\mathrm{NN}}) \rho(el,az,\hat{\alpha},\hat{\beta},\hat{\gamma})}{4\pi^2} \, \mathrm{d}\hat{\alpha} \, \mathrm{d}\hat{\gamma} \tag{14}$$

## 3.6 Radar variables derived from the database

At an elevation angle of 30° and $\hat{\beta}=0°$, Ze, ZDR, and KDP at 9.6 GHz (X-band), 35.6 GHz (Ka-band), and 94 GHz (W-band) for all particles in the database are shown in Fig. 6. A notable feature is that small ice crystals in the database can exhibit negative ZDR values, which is counter-intuitive since the particles are horizontally aligned with their largest dimension. For particles with an aspect ratio different from unity, one would generally expect a positive ZDR.

The particles showing negative ZDR are dendritic crystals with aspect ratios close to unity (0.7-0.9). Based solely on aspect ratio, their ZDR should be positive. However, the internal mass distribution is asymmetric: the vertical dimension has a higher mass density than the horizontal dimension (see examples in Fig. 1). When the aspect ratio is near unity, this density difference becomes significant, resulting in a higher vertical Ze and thus a negative ZDR. This effect is not observed for solid plates with similar size and aspect ratio because their mass distribution is nearly uniform along the vertical and horizontal axes, making aspect ratio the dominant factor for ZDR.

Comparing Ze, ZDR, and KDP at W-Band of the ice crystals in this database with those in Lu et al. (2016) (see their Fig. 4 and Fig. C1 in von Terzi et al. (2022)), we find similar features. Ice crystals with masses exceeding $10^{-8}$ kg, corresponding to a maximum dimension of approximately 3 mm (see Fig. A1), enter the non-Rayleigh scattering regime, producing a ZDR peak of 12 dB and a flattening of Ze. This behaviour is consistent with Lu et al. (2016), where dendritic crystals reach a maximum W-band ZDR of 9 dB at $D_{max}$=3 mm. Likewise, KDP in our database aligns with Lu et al. (2016), reaching 0.2 $mm°\ km-1$ at $D_{max}$=8 mm when normalized by wavelength (Fig.A1).

Aggregates in our database show some differences from those in Lu et al. (2016). At X-band, aggregates reach a maximum Ze of 27 dBz at a mass of $10^{-4}$ kg (corresponding to $D_{max} \approx 3.5$ cm), whereas in Lu et al. (2016), the maximum Ze for $D_{max} \approx 3$ cm is only 9.5 dBz. This higher reflectivity in our dataset is primarily due to rimed aggregates. Unrimed aggregates in our database reach only 5 dBz at $D_{max} \approx 3.5$ cm, comparable to Lu et al. (2016).

The transition of aggregates into the Mie-regime occurs at slightly higher masses in our dataset: around $5 \times 10^{-8}$ kg at W-band, with a pronounced flattening at $5 \times 10^{-7}$ kg ($D_{max} \approx 2$–3 mm). In contrast, Lu et al. (2016) reports the transition at $5 \times 10^{-9}$ kg. As noted in Lu et al. (2016), their aggregates enter the non-Rayleigh scattering regime earlier than those in Leinonen and Moisseev (2015), likely due to structural differences. Our aggregates consist primarily of plates, dendrites, or a mixture of dendrites and columns, whereas the Lu et al. (2016) aggregates are more akin to aggregates of needles.

ZDR values of aggregates are similar between the two databases: at W-band, most aggregates exhibit ZDR between 2–3 dB. Only high-density rimed aggregates in our dataset reach up to 5.5 dB. KDP also remains in a comparable range to Lu et al. (2016).

## 4 Database Setup

The scattering database is organized into three hierarchical levels, each corresponding to a different stage of post-processing. Due to the large size, the level 0 data is available upon request.

**Level 0** contains the raw output from the ADDA code, including the full amplitude and Mueller matrices for each particle, along with log files documenting the ADDA simulation settings and diagnostics.

In **Level 1a** these outputs are processed by summarizing the amplitude and Mueller matrix entries into `NetCDF` files for each particle type across all elevation and azimuth angles and frequencies.

**level 1b** further condenses the data into two lookup tables (LUTs), one for monomers and one for aggregates. These LUTs represent the azimuthally random orientational averages of all particles and serve as the input for the radar forward simulator, McRadar.

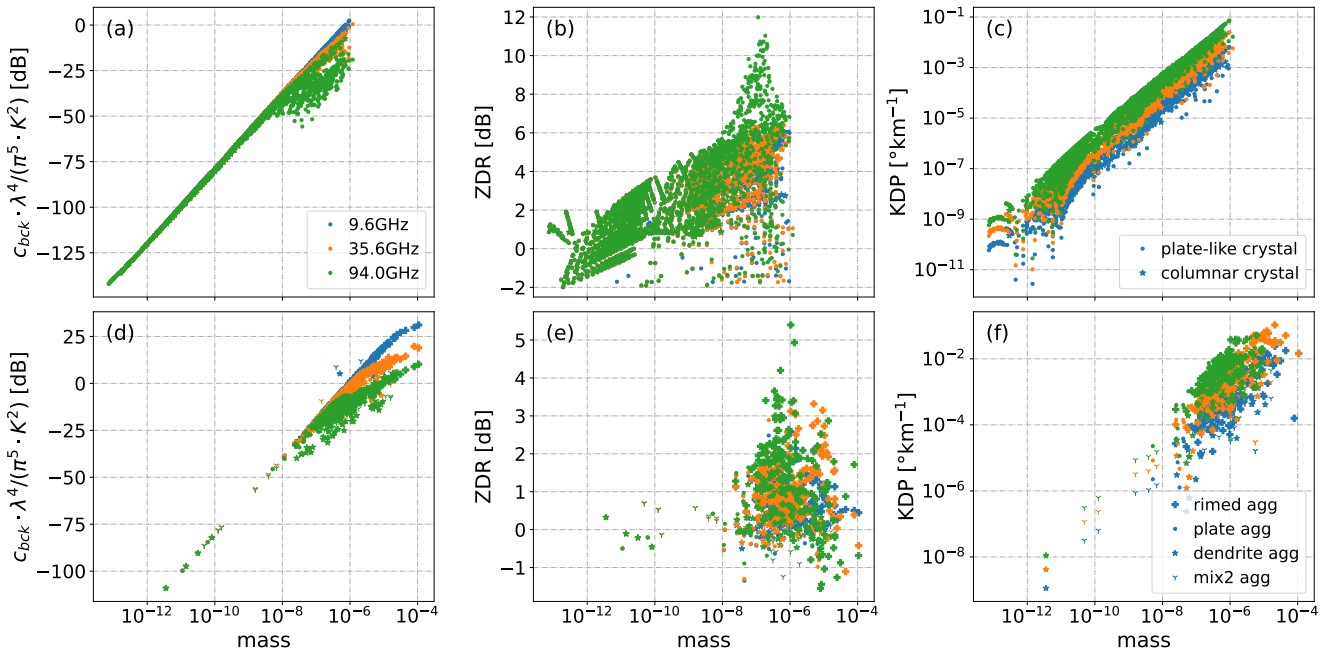

**Figure 6.** Single-particle backscattering cross sections at horizontal polarizations multiplied by $\lambda^4/(\pi^5 \cdot K^2)$ (equivalent to the single particle reflectivity) (first column), ZDR (second column), and KDP in dependency of the particle mass plotted for all particles in the database at 30° radar elevation averaged over all azimuth orientations at $\hat{\beta} = 0°$. The first row shows the scattering properties of all ice crystals at 9.6GHz (blue), 35.6GHz (orange) and 94GHz (green). The dots indicate plate-like crystals and the stars columnar crystals. The second row shows the scattering properties of all aggregates at 9.6, 35.6 and 94 GHz. The dots indicate aggregates of plates, the stars aggregates of dendrites, the trident mix2 aggregates, and the plus sign rimed aggregates.

## 4.1 Radar Simulator McRadar

McRadar is a flexible radar forward simulator designed to interface directly with the level 1b LUTs and forward simulate microphysical model output into radar observables while maintaining consistency in physical properties. Similar to existing forward operators such as the
335 Cloud-resolving model Radar SIMulator CR-SIM (Oue et al., 2020), McRadar calculates the Doppler spectrum, specific differential phase shift (KDP), and attenuation based on the microphysical inputs. Instrument-specific noise can be added to the Doppler spectra, and spectral broadening effects due to turbulence, finite beamwidth, and wind shear are optionally included following Doviak and Zrnic (1993).

McRadar also allows the user to specify particle canting angle distributions (e.g., wobbling), enabling more realistic simulations. From the Doppler spectra, standard radar moments, including equivalent reflectivity ($Z_e$), mean Doppler velocity, differential reflectivity (ZDR),
340 KDP, and linear depolarization ratio (LDR), are derived.

Allowing the ice particle to grow via deposition, during which the shape of the particle evolves using a habit prediction (Welss et al., 2024), results in a multitude of particle shapes, and therefore a large variability of particle size, mass, and aspect ratio for which scattering properties need to be found. To find the scattering properties of a particle that best fits with its microphysical properties (i.e., mass, size, and aspect ratio) to the model-predicted particle, McRadar performs a nearest-neighbour selection or regression on the level 1b LUTs, depending

on user preference. This is implemented using the `scikit-learn` Python library, which supports efficient neighbour search algorithms. By default, McRadar uses neighbours-based regression, where the output is a weighted average of the $n$ nearest neighbours based on inverse distance to the query point. This supervised learning approach follows the methodology described by Roweis et al. (2004), and more details can be found in the official documentation: https://scikit-learn.org/1.5/modules/neighbors.html.

For monomers, the scattering properties are retrieved using the nearest match in mass, $D_{\max}$, and aspect ratio in logarithmic space. For aggregates, the nearest match is found based on mass and $D_{\max}$ in logarithmic space.

### 4.1.1 Evaluation of the Nearest Neighbour Lookup

To evaluate the accuracy of the nearest-neighbour selection method, we performed a cross-validation test by randomly removing individual data points from the database. For 1,000 such points, we used the nearest-neighbour method to estimate the Mueller and amplitude matrix entries, which were then compared against the true values.

We specifically evaluated the backscattering cross sections at horizontal and vertical polarization ($c_{\mathrm{bck},hh}$ and $c_{\mathrm{bck},vv}$), ZDR, and KDP at 94 GHz and 30° elevation, using different values of $n$ (number of neighbours). As shown in Figure 7, the retrieved values exhibit a high correlation with the true values and no evident bias. Among the radar variables, ZDR shows the largest scatter.

Table 1 presents a statistical summary of the validation. Correlation coefficients for all matrix entries and radar variables range from 0.8 to 0.98, indicating high retrieval fidelity. Notably, $c_{\mathrm{bck},vv}$ has the lowest correlation, while $c_{\mathrm{bck},hh}$ shows the highest. The variability in ZDR is likely a result of the lower correlation in $c_{\mathrm{bck},vv}$. Appendix B shows the corresponding tables for the radar relevant Mueller and Amplitude matrix entries.

Because the reconstruction errors are random and unbiased, they do not accumulate in forward simulations where radar variables are obtained by considering hundreds to thousands of particles.

## 5 Application examples

This database has two key advantages: First, it allows for consistent forward simulation of the radar signatures of frozen particles with strongly varying microphysical properties. Second, the large number of particles likely reduces a bias introduced by considering a single particle to be representative of all particles with a certain size or mass. As shown in Fig. 6, especially the polarimetric scattering properties for one particle with a specific mass can span several orders of magnitude. Choosing a single particle per mass/size to represent all particles close to that mass/size proves difficult and might cause a significant bias, as was also shown in Ori et al. (2021). We want to illustrate these two advantages with two application examples which are presented in the following sections.

Both examples use the Lagrangian Monte-Carlo particle model McSnow (Brdar and Seifert, 2018). In contrast to frequently used bulk schemes, where only one or two moments of the particle size distribution are predicted, McSnow predicts the evolution of particle properties in a multidimensional phase space. The predicted particle properties include mass, density, aspect ratio, number of monomers per aggregate, or rime mass. The properties are predicted for so-called super-particles, which evolve through depositional growth, aggregation, riming, secondary ice production, melting, and warm-rain processes such as drop breakup (Brdar and Seifert, 2018; Bringi et al., 2020).

### 5.1 Simulated radar quantities of single growing ice crystals

Showing the flexibility of the DDA database in terms of large variability of particle properties is done by forward simulating the trajectories (or growth histories) of ice crystals that are sedimenting towards the ground while growing through deposition. The habit of ice crystals

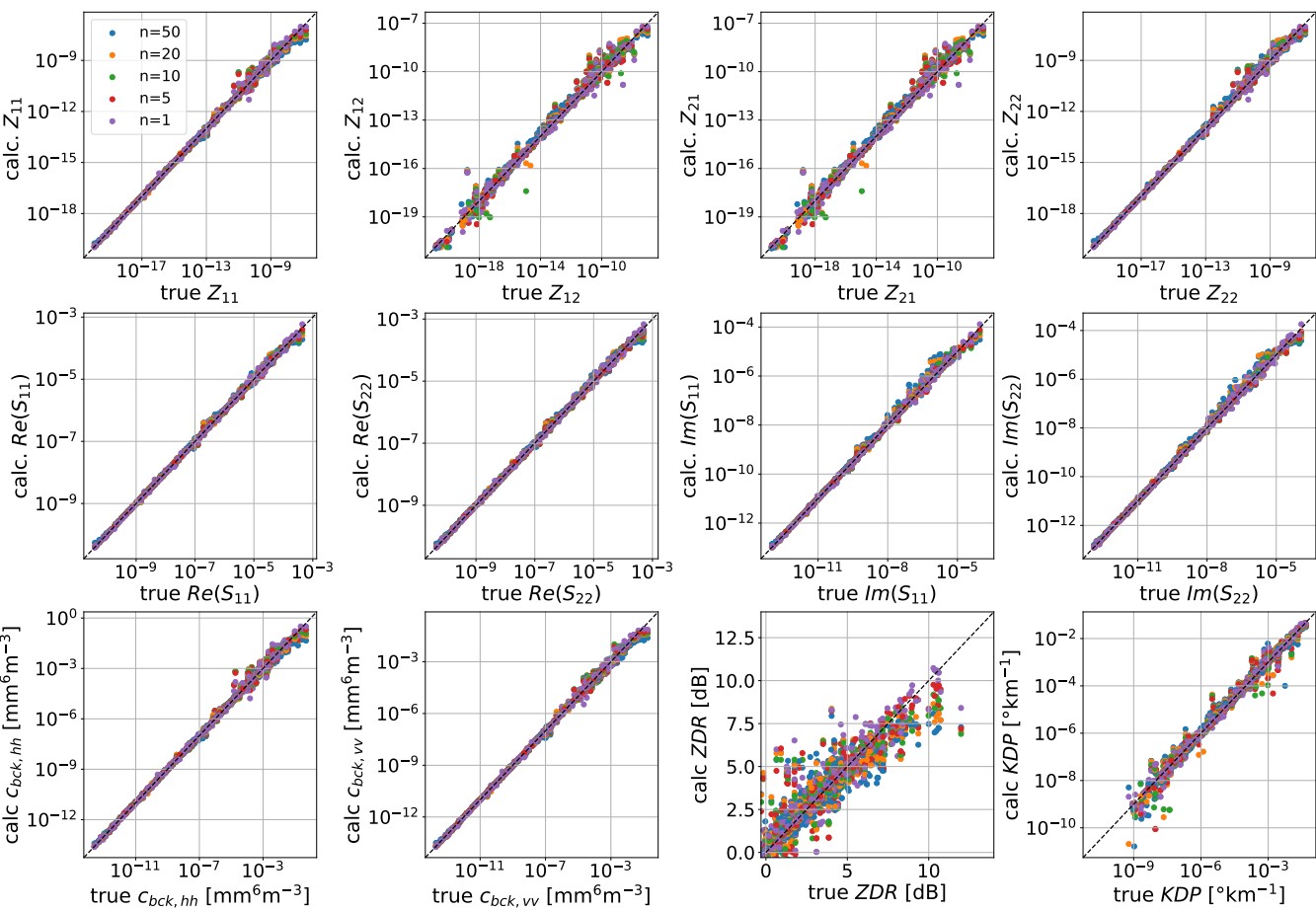

**Figure 7.** Result of the validation of the nearest neighbour selection from the scattering database. Shown are scatter plots of the true values and calculated values at 94GHz and 30° elevation. The first row shows the entries of the Mueller matrix, the second row of the Amplitude matrix, the third row of the calculated reflectivity at horizontal and vertical polarization, ZDR, and KDP.

strongly depends on the ambient temperature and super-saturation (i.e. Bailey and Hallett, 2009). Recently, a habit prediction has been
380   implemented in McSnow (Welss et al., 2024). The habit of an ice crystal can be described by its aspect ratio $\phi$. In McSnow, $\phi$ is defined as the ratio of the $c$ to the $a$-axis $\phi = \frac{c}{a}$, where the a-axis is the maximum dimension of the basal face, and the $c$-axis is the maximum dimension of the prism face. The change of $\phi$ can be calculated with

$$\frac{\mathrm{d}c}{\mathrm{d}a} = \mathrm{d}\phi = \frac{\alpha_\mathrm{c}(T)}{\alpha_\mathrm{a}(T)}\phi = \Gamma(T)\phi \qquad (15)$$

with the deposition coefficients of axis $c$ ($\alpha_c$) and axis $a$ ($\alpha_a$) and the temperature (T) dependent inherent growth function $\Gamma$. $\Gamma$ has been
385   determined by different studies through comparison with observational data as well as laboratory experiments (i.e. Welss et al., 2024; Chen and Lamb, 1994). The change of mass due to depositional growth can then be calculated with the well-known mass growth equation (i.e.

| | $n$ | 1 | 5 | 10 | 20 | 50 |
|---|---|---|---|---|---|---|
| $c_{bck,hh}$ [mm$^6$m$^{-3}$] | mean error | $9.60 \times 10^{-4}$ | $9.25 \times 10^{-4}$ | $1.06 \times 10^{-3}$ | $1.24 \times 10^{-3}$ | $1.45 \times 10^{-3}$ |
| | median error | $3.99 \times 10^{-10}$ | $2.50 \times 10^{-10}$ | $3.40 \times 10^{-10}$ | $3.13 \times 10^{-10}$ | $4.77 \times 10^{-10}$ |
| | correlation coefficient | 0.94 | 0.91 | 0.90 | 0.87 | 0.84 |
| | bias | $1.97 \times 10^{-4}$ | $2.58 \times 10^{-4}$ | $3.36 \times 10^{-4}$ | $3.72 \times 10^{-4}$ | $4.14 \times 10^{-4}$ |
| $c_{bck,vv}$ [mm$^6$m$^{-3}$] | mean error | $6.91 \times 10^{-4}$ | $6.26 \times 10{-4}$ | $7.38 \times 10^{-4}$ | $7.58 \times 10^{-4}$ | $8.28 \times 10^{-4}$ |
| | median error | $2.06 \times 10^{-10}$ | $1.11 \times 10^{-10}$ | $8.39 \times 10^{-11}$ | $1.28 \times 10^{-10}$ | $1.63 \times 10^{-10}$ |
| | correlation coefficient | 0.84 | 0.84 | 0.79 | 0.79 | 0.77 |
| | bias | $7.09 \times 10^{-5}$ | $8.57 \times 10^{-5}$ | $9.57 \times 10^{-5}$ | $1.02 \times 10^{-4}$ | $1.16 \times 10^{-4}$ |
| $ZDR$ [ dB] | mean error | 0.35 | 0.43 | 0.52 | 0.59 | 0.72 |
| | median error | 0.11 | 0.16 | 0.21 | 0.26 | 0.36 |
| | correlation coefficient | 0.94 | 0.93 | 0.91 | 0.90 | 0.88 |
| | bias | $-0.25$ | $-0.84$ | $-1.20$ | $-1.20$ | $-1.13$ |
| $KDP$ [°km$^{-1}$] | mean error | $2.18 \times 10^{-4}$ | $2.40 \times 10^{-4}$ | $2.72 \times 10^{-4}$ | $3.23 \times 10^{-4}$ | $3.84 \times 10^{-4}$ |
| | median error | $7.31 \times 10^{-8}$ | $6.28 \times 10^{-8}$ | $8.47 \times 10^{-8}$ | $1.05 \times 10^{-7}$ | $1.63 \times 10^{-7}$ |
| | correlation coefficient | 0.97 | 0.97 | 0.97 | 0.96 | 0.96 |
| | bias | $-8.91 \times 10^{-7}$ | $1.80 \times 10^{-6}$ | $3.37 \times 10^{-6}$ | $4.52 \times 10^{-6}$ | $6.44 \times 10^{-6}$ |

**Table 1.** The mean error, median error, standard deviation of the error as well as the correlation and bias between the true backscattering cross sections at horizontal and vertical polarization, the true ZDR, the true KDP and the calculated backscattering cross sections, ZDR and KDP.

Lohmann et al., 2016, Chapter 7)

$$\frac{\mathrm{d}m}{\mathrm{d}t} = 4\pi C D_\mathrm{v} \bar{f} \frac{S_i - 1}{F_\mathrm{k}^i + F_\mathrm{d}^i} \tag{16}$$

with the mass of the particle $m$, the water vapour diffusivity $D_v$, ambient saturation ratio with respect to ice $S_i$, particle ventilation coefficient $\bar{f}$. $F_\mathrm{k}^i$, describes the latent heat release due to sublimation and the diffusion of heat away from the ice particle, and $F_\mathrm{d}^i$ describes the diffusion of water vapour towards the ice particle. They are expressed as

$$F_\mathrm{k}^i = \left( \frac{L_\mathrm{s}}{R_\mathrm{v} T} - 1 \right); \quad F_\mathrm{d}^i = \frac{R_\mathrm{v} T}{D_\mathrm{v} e_\mathrm{s,i}(T)} \tag{17}$$

With $L_\mathrm{s}$ the latent heat of sublimation, $T$ the ambient temperature, $R_\mathrm{v}$ the gas constant of water vapour and $e_\mathrm{s,i}(T)$ the saturation vapour pressure above ice at the temperature $T$.

The change of $\phi$ is taken into account through the capacity $C$ (with subscript p for plate-like particles, c for columnar particles):

$$C_p = \frac{\alpha \epsilon_\mathrm{p}}{\sin^{-1}(\epsilon_\mathrm{p})}; \quad C_c = \frac{c \epsilon_\mathrm{c}}{\ln(1 + \epsilon_c)\phi} \tag{18}$$

with $\epsilon_\mathrm{p} = \sqrt{1 - \phi^2}$ and $\epsilon_\mathrm{c} = \sqrt{1 - \phi^{-2}}$.

An additional variable necessary to determine the habit of an ice crystal is its overall density. This density is, for example, reduced from solid ice density through branching of dendrites at temperatures close to $-15°C$ or hollowing of columns at temperatures between $-5$ and

$-10°C$. In McSnow, the volume enveloping ice crystals is simulated as a spheroid, which does not allow to simulate branching or hollowing directly. Instead, the density of the spheroid is adapted by amplifying the increase of the ice volume $V$ relative to the mass $m$.

$$\frac{\mathrm{d}V}{\mathrm{d}t} = \frac{1}{\rho_{\mathrm{depo}}} \frac{\mathrm{d}m}{\mathrm{d}t} \tag{19}$$

The deposition density $\rho_{depo}$ depends on the ambient temperature and is proportional to $\rho_i \Gamma(T)$ for plate-like particles and $\rho_i \Gamma(T)^{-1}$ for columnar particles with ice density $\rho_i$. For further details about the habit prediction, the reader is referred to Welss et al. (2024).

For this application, the modified inherent growth function (IGF2) from Welss et al. (2024) has been used. While McSnow is a Lagrangian Monte-Carlo particle model, designed to simulate populations of particles, it is also possible to simulate the evolution of single ice particles due to various microphysical processes, in this case depositional growth and sedimentation. Keeping the ambient super-saturation with respect to ice constant at 5%, setting the temperature at the ground to 0°C and assuming a constant lapse rate of 6 K km$^{-1}$, 5 particles were simulated starting at different heights (temperatures). The particles were initialized with a diameter of 10 $\mu$m and an aspect ratio of unity. During the simulation, the particles grow by deposition as predicted by the habit prediction while sedimenting through different temperature regimes towards the ground. No interactions between the particles have been assumed.

Depending on the temperature at which a particle was initialized, the physical properties vary significantly (see Fig. 8). The particles that started at temperatures colder than -22°C are growing columnar (with an aspect ratio larger than 1), while the particles initialized at temperatures warmer than -22°C grow plate-like (with an aspect ratio smaller than 1). The particle that was initialized at -15°C experiences the strongest depositional growth, reaching a maximum dimension (D$_{max}$) of 3.5 mm and an aspect ratio of 0.0015, consistent with laboratory studies (Pruppacher and Klett, 1997). The density of this particle is significantly reduced, reaching a minimum of 390 kg m$^{-3}$. This indicates significant branching, as is expected by the predominantly dendritic growth at -15°C. The smallest mass and D$_{max}$ is reached by the particle nucleated at -10°C. This particle initially grows plate-like, but is sedimenting out of the plate growth region into the columnar growth region at temperatures warmer than -10°C. Therefore, the particle only slowly increases its mass and D$_{max}$ while at the same time increasing its aspect ratio toward unity.

This large variability of ice microphysical properties produced by the McSnow simulation highlights the need for a scattering database consisting of ice particles with a similar variability. For the database in this study, we therefore generated specific ice crystals that can provide this large variability in ice microphysical properties. The corresponding forward simulations using the nearest-neighbour regression with $n = 10$ to access the DDA LUTs, are shown in the second and third row of Fig. 8.

The most interesting scattering properties are expected from particles nucleated within the dendritic growth zone (between -10 and -20°C). Those particles experience the strongest depositional growth, and reduce their density through branching. While growth effects such as branching can be seen in a reduction of ZDR, e.g. at -14.2°C for the particle nucleated at -15°C, also differences at different frequencies can be observed. While at X-Band, ZDR of the particle nucleated at -15°C only slightly increases towards warmer temperatures, at W-Band a strong peak at -8°C is observed, where a ZDR of 14.5 dB is reached. Looking at the single particle scattering properties in Figure 6, it is evident that particles with a mass between $10^{-8}$ and $10^{-6}$ kg are producing large ZDR at W-Band. This mass corresponds to approximately a D$_{max}$ of 3 mm, which is close to the wavelength of W-band, causing non-Rayleigh scattering effects. While both ZeH and ZeV are in the non-Rayleigh regime (ZeH and ZeV are not increasing as fast at W-Band as at X-Band), ZeV is affected stronger than ZeH, causing the large increase in ZDR.

In-situ observations sometimes reveal particles which must have grown in multiple growth regimes (Bailey and Hallett, 2009). For example, capped-columns are observed when an ice crystal initially grows columnar, then sediments into the plate-like growth regime and develops plates on both ends of the column. In this simulation, the particle nucleated at -25°C could potentially develop as a capped

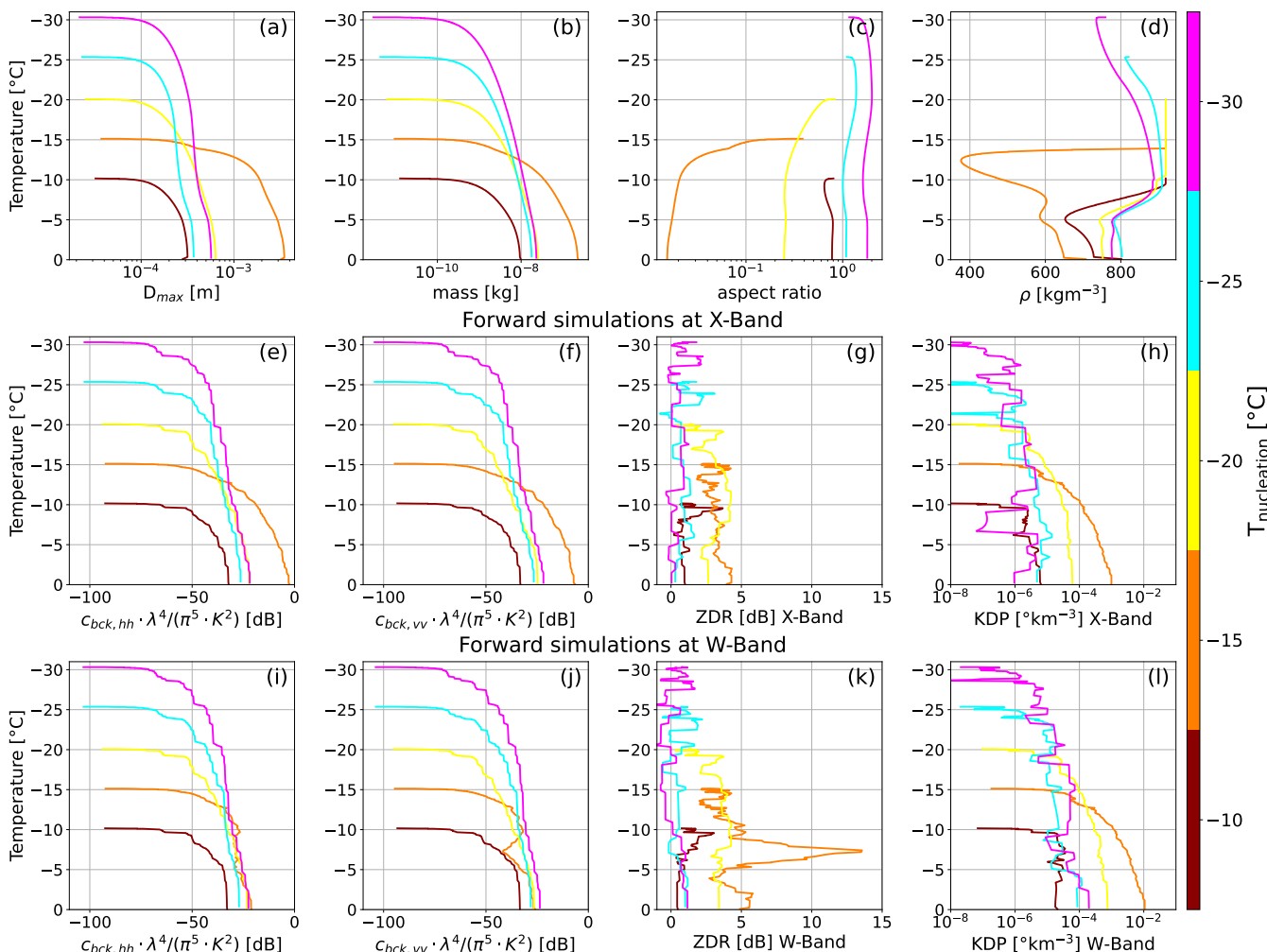

**Figure 8.** First row: Microphysical properties of single ice crystals nucleated at different temperatures and growing by deposition using the habit prediction (Welss et al., 2024) while sedimenting towards the ground. Shown are the evolution of the maximum dimension $D_{max}$ (a), mass (b), aspect ratio (c) and density $\rho$ (d) with ambient temperature. The second row shows the corresponding radar forward simulations for each particle at 9.6 GHz (X-Band) and 30° elevation. Shown are the backscattering cross section at horizontal (e) and vertical (f) polarizations multiplied by $\lambda^4/(\pi^5 \cdot K^2)$ (equivalent to the single particle reflectivity), the single particle ZDR (g), and the single particle KDP (h). The third row shows the same radar forward simulations at 94 GHz (W-Band) and 30° elevation.

column in nature. In the model, these kinds of particles are, however, not represented. Only the change in aspect ratio and density of the particle can be captured. The transition from the columnar to the plate-like growth regime causes a reduction of the aspect ratio towards unity. While the addition of capped-columns is in principle possible, we have decided not to include these particles in the scattering database, as this introduces the need to include the growth history of the ice particle in the forward simulations. Rather, in the database, we have generated plates, dendrites, and columns with aspect ratios close to unity. In Bailey and Hallett (2009), it has also been shown in a laboratory study,


that when a column transitions into the plate-like growth regime, under certain circumstances it can also just reduce its aspect ratio rather than producing a plate at each end, which might justify our approach.

## 5.2 Simulated radar signatures of idealized 1D McSnow simulation

Because forward simulations rely on a finite set of particle geometries, they are inevitably affected by sampling variability, i.e., uncertainty and potentially biases arising from the limited number of shapes used to represent the full natural population. The Lu et al. (2016) database includes 1000 particles, a significantly larger number of particles as compared to the Brath et al. (2020) database, thus allowing for more variability in particle properties. However, this variability is still not large enough to cover the variability of ice particle properties in McSnow simulations, especially the aspect ratios lack in variability compared to our database (see Figure C1). Compared to our database, the Lu et al.
(2016) database contains approximately 1/3 particles, thus still potentially introducing a bias by not considering enough particles. The impact is likely two-fold: first, by using a smaller number of particles, the microphysical properties (mass, Dmax, aspect ratio) of the simulated particle might not be matched well enough, and second, considering a smaller number increases the statistical noise introduced by picking a random particle.

To investigate potential biases arising from particle representation, we performed an idealized one-dimensional simulation using McSnow.
McSnow is particularly well-suited for this purpose, as it explicitly simulates the microphysical evolution of individual particles. For this case study, we adopt the same atmospheric setup as in Section 5.1: a surface temperature of 0°C, a lapse rate of 6 K km$^{-1}$, and a constant relative humidity with respect to ice of 105%. Ice particles are initialized in a nucleation layer spanning 2–6 km altitude, corresponding to temperatures from –10°C to –30°C. In this layer, we assume a constant nucleation rate of 0.375 particles s$^{-1}$m$^{-3}$, leading to a maximum concentration of 2000 particles m$^{-3}$ at -15°C.

During the simulation, ice particles may grow by deposition and aggregation. Since we use the temperature-dependent sticking efficiency from Connolly et al. (2012), aggregation becomes an important growth mechanism only at temperatures warmer than -20°C. In the current McSnow setup, aggregates follow predefined mass–size and area–size relationships; here, we assume aggregates composed of side planes, columns, and bullet rosettes as described in Mitchell et al. (1996).

For our reference radar forward simulation, we use all particles of our database, and $n = 10$ nearest neighbours to estimate the scattering
properties of each simulated super-particle. To show the benefit of our database, which uses approximately 3 times more particles as the Lu et al. (2016) database, we then randomly downsampled our database to contain only 1/3 of particles. Assuming a homogeneous coverage in the microphysical space, in order to have a similar look-up radius we reduced the number of nearest neighbours considered for the look-up to $n = 3$. We then forward simulated the McSnow output. In total, we performed the random downsampling and subsequent forward simulations 100 times.

### 5.2.1 Microphysical evolution


The idealized McSnow simulation (Figure 9) exhibits radar signatures commonly observed in the field, as reported for example by von Terzi et al. (2022) and references therein. A continuous increase in Ze toward the ground, accompanied by a rise in the dual-wavelength ratio (DWR), is a strong indicator of intensifying aggregation. Also in the Doppler spectra, aggregation leads to a rapidly increasing spectral Ze at a temperature close to -15°C. The aggregation of fast-falling columnar ice crystals leads to a lower fall velocity of the resulting aggregates
with larger reflectivities. A second mode in the spectrum in this temperature region can be explained by nucleation of ice particles, which grow into dendrites that have low fall velocities due to their large cross-sectional area. Together, these processes cause the prominent dip in the mean Doppler velocity (MDV) close to -15°C.

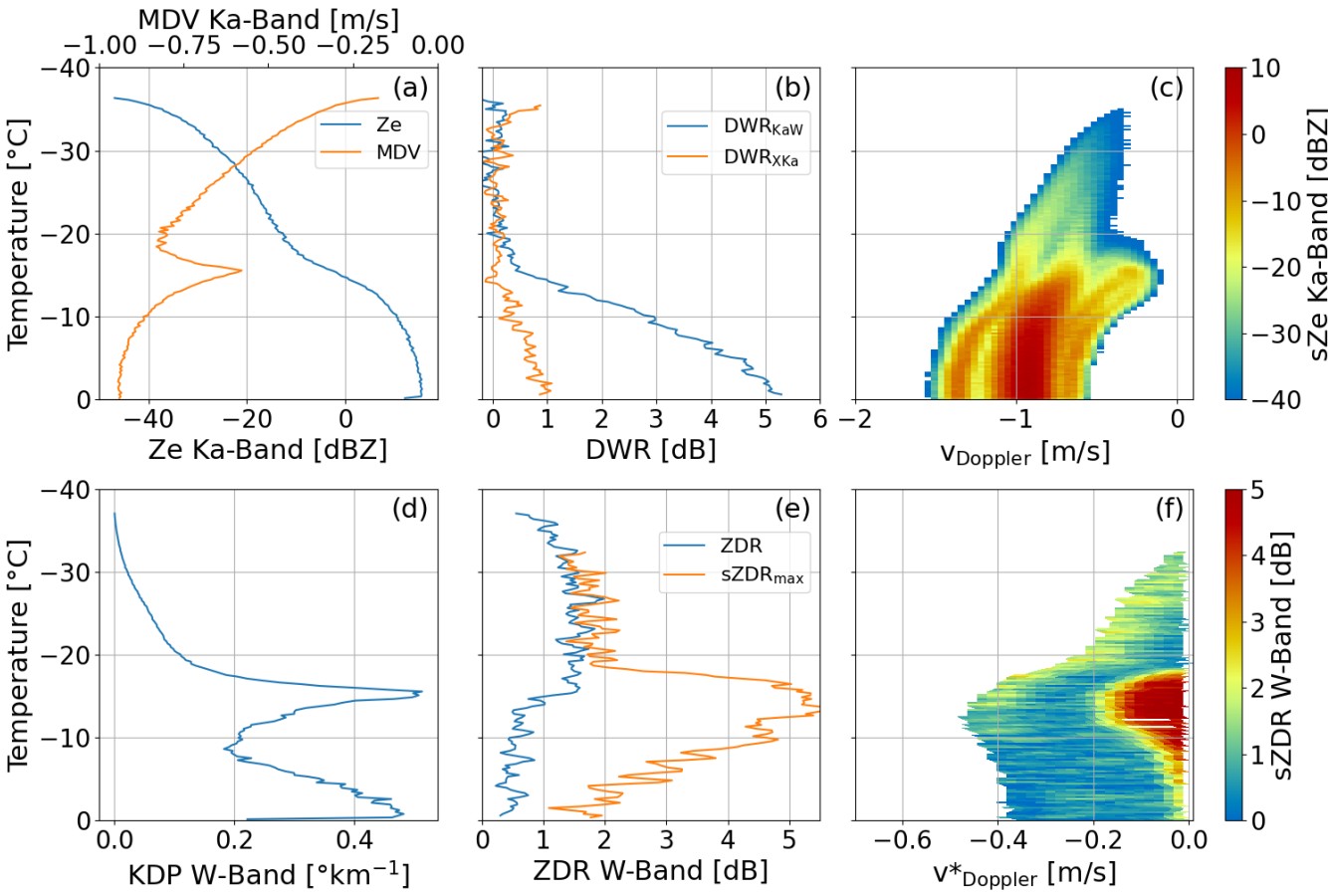

**Figure 9.** Forward simulation of the 1D McSnow simulation. Shown are in the first row: Ze and MDV at Ka-Band (a), $DWR_{KaW}$ and $DWR_{XKa}$ (b) and the spectral Ze from Ka-Band (c), all at 90° elevation. The second row contains the forward simulated polarimetric variables at 30° elevation and W-Band: KDP (d), ZDR and $sZDR_{max}$ (e) and spectral ZDR (d).

The dendritic ice crystals also produce characteristic polarimetric signatures. KDP begins to increase slightly at around -30°C, followed by a strong increase at -18°C, reaching a maximum of 0.55 $^{\circ}$ km$^{-1}$ in the dendritic growth layer at -15°C. Also at -18°C, $sZDR_{max}$ begins to increase, reaching values up to 5 dB at -15°C. Below this layer, toward the ground, all polarimetric variables decrease slightly, reflecting the aggregation of the largest dendritic ice crystals.

### 5.2.2 Investigating the impact of smaller number of particles in the scattering database

Using a finite number of particle geometries in the forward simulations inevitably introduces statistical sampling noise. Since our database contains the largest number of ice particles (approximately three times larger than the Lu et al. (2016) database), we are able to quantify this sampling uncertainty.

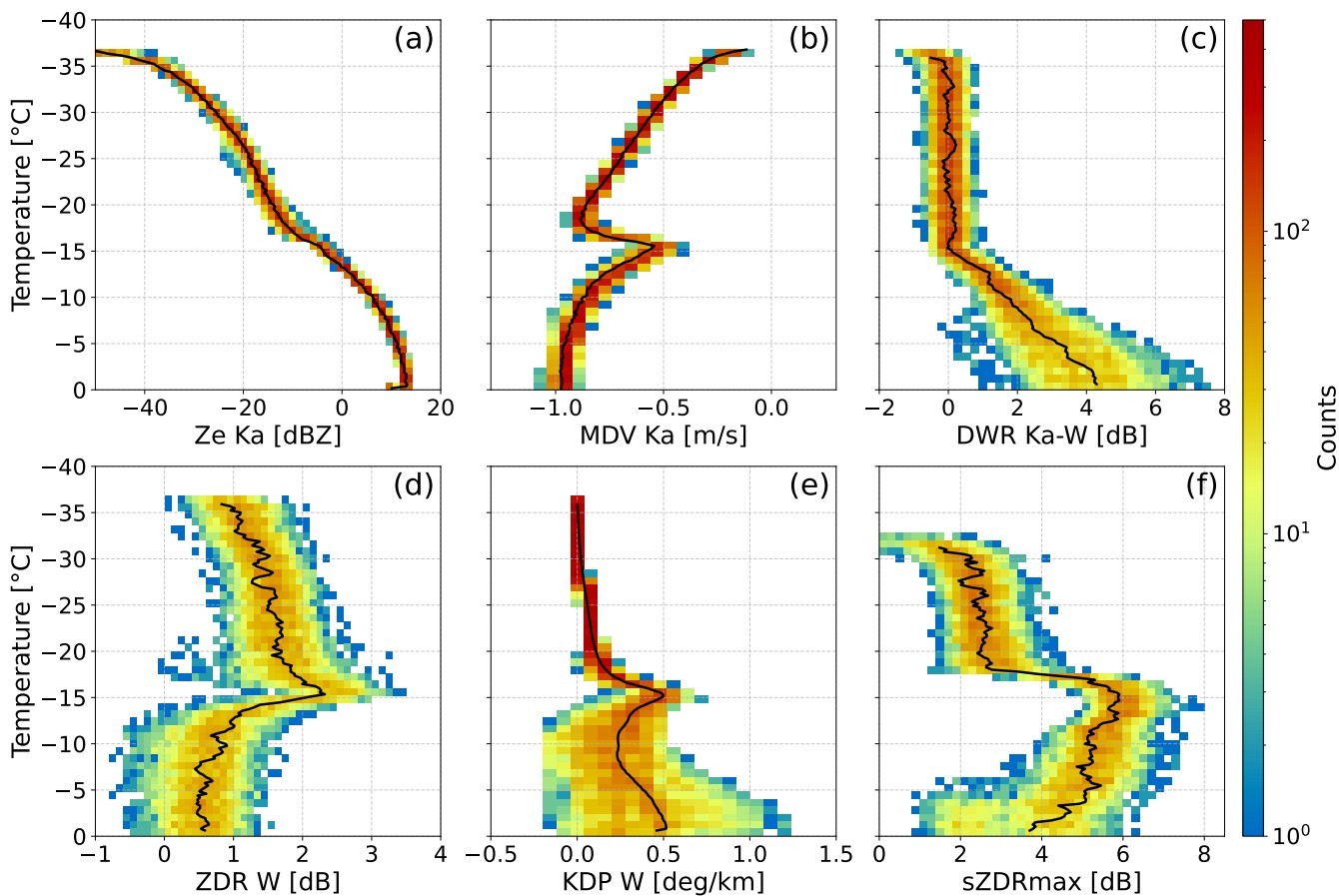

**Figure 10.** Variability of scattering properties if only $1/3$ of particles from the database is used to forward simulate the McSnow simulation. Shown are Ze (a), MDV (b), DWR$_{KaW}$ (c), ZDR (d), KDP (e) and sZDRmax (f). The black line indicates the original simulation, where the entire database was used.

The impact of reduced sample size depends strongly on the radar variable, shown in Figure 10. For Ze and MDV, sampling noise is surprisingly small: the resulting distributions remain narrow, and the differences relative to the full-database simulation are minor. This indicates that these bulk variables are relatively insensitive to the precise set of particle shapes used.

In contrast, for the dual-wavelength ratio (DWR), the spread increases significantly toward warmer temperatures; depending on the random
subset, simulated DWR values range from roughly 2 to 8 dB, demonstrating that this variable is highly sensitive to the morphological diversity of the particles. This behaviour further underlines the work previously done by (e.g. Mason et al., 2019), who have shown that DWR$_{KaW}$ and DWR$_{XKa}$ significantly vary with the particles internal structure.

Variables that respond directly to particle asymmetry exhibit large variability under reduced sampling. ZDR varies by approximately $\pm 1$ dB relative to the reference simulation, and the spread in KDP grows with temperature, reaching its maximum near the melting layer,
where values between $-0.1$ and $1.2°$ km$^{-1}$ are possible.

Overall, this experiment highlights that not all radar observables are equally affected by uncertainties in particle-shape sampling. While bulk quantities (e.g., Ze, MDV) are comparatively robust, polarimetric and multi-frequency variables are highly sensitive to the underlying particle geometry. Although our database is large, it cannot fully capture the true natural variability of snowflakes. Increasing the number and diversity of simulated particle shapes would likely reduce sampling noise further and help determine whether these sensitivities converge with larger ensembles.

## 6 Summary and conclusions

New developments in microphysical models, such as Lagrangian particle models including a habit prediction for ice particles or the P3 scheme (Morrison et al., 2025), which continuously evolves the microphysical properties of ice particles, necessitate a scattering database that can cope with largely varying microphysical properties. In this study, we have presented a new DDA scattering database consisting of 2600 ice crystals, including a large variation of dendritic ice crystals, plates, columns, and needles, as well as 450 aggregates of plates, dendrites, mixed monomer habits, and various degrees of riming. The scattering properties of these particles at C-, X-, Ka-, and W-Band at various elevation and azimuth angles, as well as orientationally averaged scattering properties, are available. The database is structured on three levels. Level 0 comprises the raw ADDA output (including the Mueller and Amplitude matrices) for each wavelength, elevation, and azimuth orientation. level 1a summarizes the for radar applications relevant Mueller and Amplitude matrix entries for each particle type in netcdf files. level 1b consists of the averages of azimuthally random orientations of all particles summarized in LUTs. These LUTs can be accessed via the radar simulator McRadar, where radar Doppler spectra at horizontal and vertical polarization as well as KDP are simulated, or interfaced to other forward operators.

In this study, we have demonstrated the two main advantages of this database: First, due to the large variety of realistic ice crystals available, this database can be utilized to forward simulate complex model output, where the ice particle habit is predicted. This enables us to study depositional growth signatures of ice crystals that could be expected, for example, in the dendritic growth layer. Second, using realistically shaped ice particles in forward simulations always poses the challenge of choosing the "correct" shape of ice particle. This shape has a significant influence on the scattering properties and can thus cause a bias in forward simulations (Ori et al., 2021). This database is based on a large number of ice particles (i.e. approximately three times more particles than the Lu et al. (2016) database), which likely reduces the bias of choosing an arbitrary shape.

While the focus of the database thus far was placed on dendritic ice crystals, current developments in McSnow shift our focus to aggregate properties. Therefore, in the near future, the database will be expanded by the scattering properties of a large ensemble of aggregates.

*Code and data availability.* The paper presents the new DDA database, level 1a and level 1b, which are archived on zenodo under the terms of the Creative Commons Attribution 4.0 Internationals licence (https://doi.org/10.5281/zenodo.16792943, von Terzi et al., 2025). The level 0 data are also available under the Creative Commons Attribution 4.0 Internationals licence (https://doi.org/10.57970/cgtek-fen77, von Terzi, 2025). This study introduced the radar forward operator McSnow, which is released under the terms of the GNU Public License version 3 and available at https://github.com/lterzi/McRadar. The exact version used in this publication is archived on zenodo (Leonie von Terzi, 2025). All codes used to generate the ice crystals, calculate the scattering properties and produce the Figures can be found in https://github.com/lterzi/DDA_database_gmd

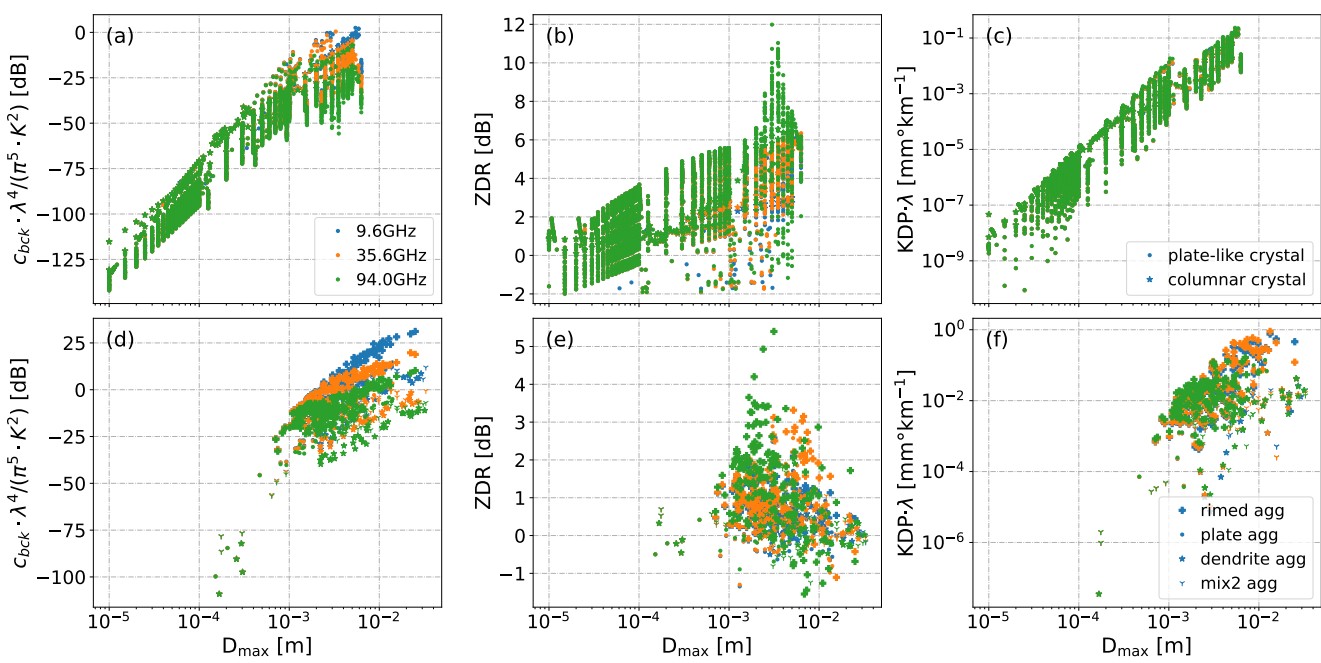

**Figure A1.** As Figure 6, but in dependency of the maximum dimension $D_{\mathrm{max}}$ instead of mass. KDP is normalized with the wavelength to allow an easier comparison with Figure C1 of von Terzi et al. (2022)

Figure A1 shows the scattering properties of all particles in the database at an elevation of 30° and particle canting angle $\beta$ of 0°. This figure allows a more direct comparison to other studies such as Lu et al. (2016) or von Terzi et al. (2022).

**Appendix B:  Uncertainty analysis of the selection from the Look-up tables**

In this section, additional information about the evaluation of the nearest neighbour look-up (Section 4.1.1) is provided. Namely, the statistical
quantities (error, correlation coefficient and bias) of the Mueller matrix entries (Table B1) and the Amplitude matrix (Table B1) are provided.

|  | $n_{neighbours}$ | 5 | 10 | 20 | 50 |
|---|---|---|---|---|---|
| $Z_{11}$ | mean error | $2.13 \times 10^{-4}$ | $2.37 \times 10^{-4}$ | $2.46 \times 10^{-4}$ | $3.05 \times 10^{-4}$ |
|  | median error | $3.70 \times 10^{-11}$ | $5.79 \times 10^{-11}$ | $4.48 \times 10^{-11}$ | $5.54 \times 10^{-11}$ |
|  | correlation coefficient | 0.89 | 0.77 | 0.75 | 0.73 |
|  | bias | $2.02 \times 10^{-5}$ | $2.25 \times 10^{-5}$ | $2.41 \times 10^{-5}$ | $2.72 \times 10^{-5}$ |
| $Z_{12}$ | mean error | $1.08 \times 10^{-4}$ | $1.26 \times 10^{-4}$ | $1.40 \times 10^{-4}$ | $1.61 \times 10^{-4}$ |
|  | median error | $3.08 \times 10^{-11}$ | $3.71 \times 10^{-11}$ | $4.06 \times 10^{-11}0$ | $6.11 \times 10^{-11}$ |
|  | correlation coefficient | 0.98 | 0.97 | 0.95 | 0.87 |
|  | bias | $7.95 \times 10^{-6}$ | $8.80 \times 10^{-6}$ | $9.35 \times 10^{-6}$ | $9.74 \times 10^{-6}$ |
| $Z_{21}$ | mean error | $1.08 \times 10^{-4}$ | $1.26 \times 10^{-4}$ | $1.40 \times 10^{-4}$ | $1.61 \times 10^{-4}$ |
|  | median error | $3.08 \times 10^{-11}$ | $3.71 \times 10^{-11}$ | $4.06 \times 10^{-11}$ | $6.11 \times 10^{-11}$ |
|  | correlation coefficient | 0.98 | 0.97 | 0.95 | 0.87 |
|  | bias | $7.95 \times 10^{-6}$ | $8.80 \times 10^{-6}$ | $9.35 \times 10^{-6}$ | $9.74 \times 10^{-6}$ |
| $Z_{22}$ | mean error | $2.13 \times 10^{-4}$ | $2.37 \times 10^{-4}$ | $2.46 \times 10^{-4}$ | $3.05 \times 10^{-4}$ |
|  | median error | $3.70 \times 10^{-11}$ | $5.79 \times 10^{-11}$ | $4.48 \times 10^{-11}$ | $5.54 \times 10^{-11}$ |
|  | correlation coefficient | 0.89 | 0.77 | 0.75 | 0.73 |
|  | bias | $2.02 \times 10^{-5}$ | $2.25 \times 10^{-5}$ | $2.41 \times 10^{-5}$ | $2.72 \times 10^{-5}$ |

**Table B1.** Same as Table 1 but for the entries of the Mueller Matrix relevant for radar applications

|  | $n_{neighbours}$ | 5 | 10 | 20 | 50 |
|---|---|---|---|---|---|
| $Re(S_{11})$ | mean error | $1.04 \times 10^{-3}$ | $1.23 \times 10^{-3}$ | $1.63 \times 10^{-3}$ | $2.37 \times 10^{-3}$ |
|  | median error | $1.63 \times 10^{-7}$ | $1.65 \times 10^{-7}$ | $1.56 \times 10^{-7}$ | $2.23 \times 10^{-7}$ |
|  | correlation coefficient | 0.99 | 0.98 | 0.97 | 0.94 |
|  | bias | $4.06 \times 10^{-4}$ | $5.20 \times 10^{-4}$ | $7.16 \times 10^{-4}$ | $9.76 \times 10^{-4}$ |
| $Re(S_{22})$ | mean error | $1.91 \times 10^{-3}$ | $2.31 \times 10^{-3}$ | $2.99 \times 10^{-3}$ | $3.81 \times 10^{-3}$ |
|  | median error | $2.52 \times 10^{-7}$ | $3.82 \times 10^{-7}$ | $5.39 \times 10^{-7}$ | $6.32 \times 10^{-7}$ |
|  | correlation coefficient | 0.99 | 0.98 | 0.97 | 0.95 |
|  | bias | $7.31 \times 10^{-4}$ | $1.01 \times 10^{-3}$ | $1.26 \times 10^{-3}$ | $1.83 \times 10^{-3}$ |
| $Im(S_{11})$ | mean error | $1.93 \times 10^{-4}$ | $2.05 \times 10^{-4}$ | $2.47 \times 10^{-4}$ | $3.62 \times 10^{-4}$ |
|  | median error | $5.47 \times 10^{-10}$ | $5.28 \times 10^{-10}$ | $4.57 \times 10^{-10}$ | $6.56 \times 10^{-10}$ |
|  | correlation coefficient | 0.94 | 0.93 | 0.91 | 0.86 |
|  | bias | $1.63 \times 10^{-5}$ | $1.92 \times 10^{-5}$ | $2.44 \times 10^{-5}$ | $2.95 \times 10^{-5}$ |
| $Im(S_{22})$ | mean error | $4.29 \times 10^{-4}$ | $4.87 \times 10^{-4}$ | $6.23 \times 10^{-4}$ | $7.62 \times 10^{-4}$ |
|  | median error | $1.13 \times 10^{-9}$ | $1.74 \times 10^{-9}$ | $1.98 \times 10^{-9}$ | $2.85 \times 10^{-9}$ |
|  | correlation coefficient | 0.94 | 0.92 | 0.89 | 0.84 |
|  | bias | $4.39 \times 10^{-5}$ | $5.32 \times 10^{-5}$ | $6.08 \times 10^{-5}$ | $7.24 \times 10^{-5}$ |

**Table B2.** Same as Table 1 but for the entries of the Amplitude matrix relevant for radar applications

## Appendix C: Microphysical properties of particles in the Lu et al. (2016) database

Figure C1 compares the microphysical properties (mass , $D_{max}$ and aspect ratio) of the Lu et al. (2016) database with the microphysical properties simulated in McSnow (see Figure 9). While the Lu et al. (2016) database already provides a large set of ice crystals, it is not

enough to cover the large space of microphysical properties simulated with McSnow. In contrast, the database presented in this study is capable to span the large microphysical space provided by McSnow (second row of Figure C1).

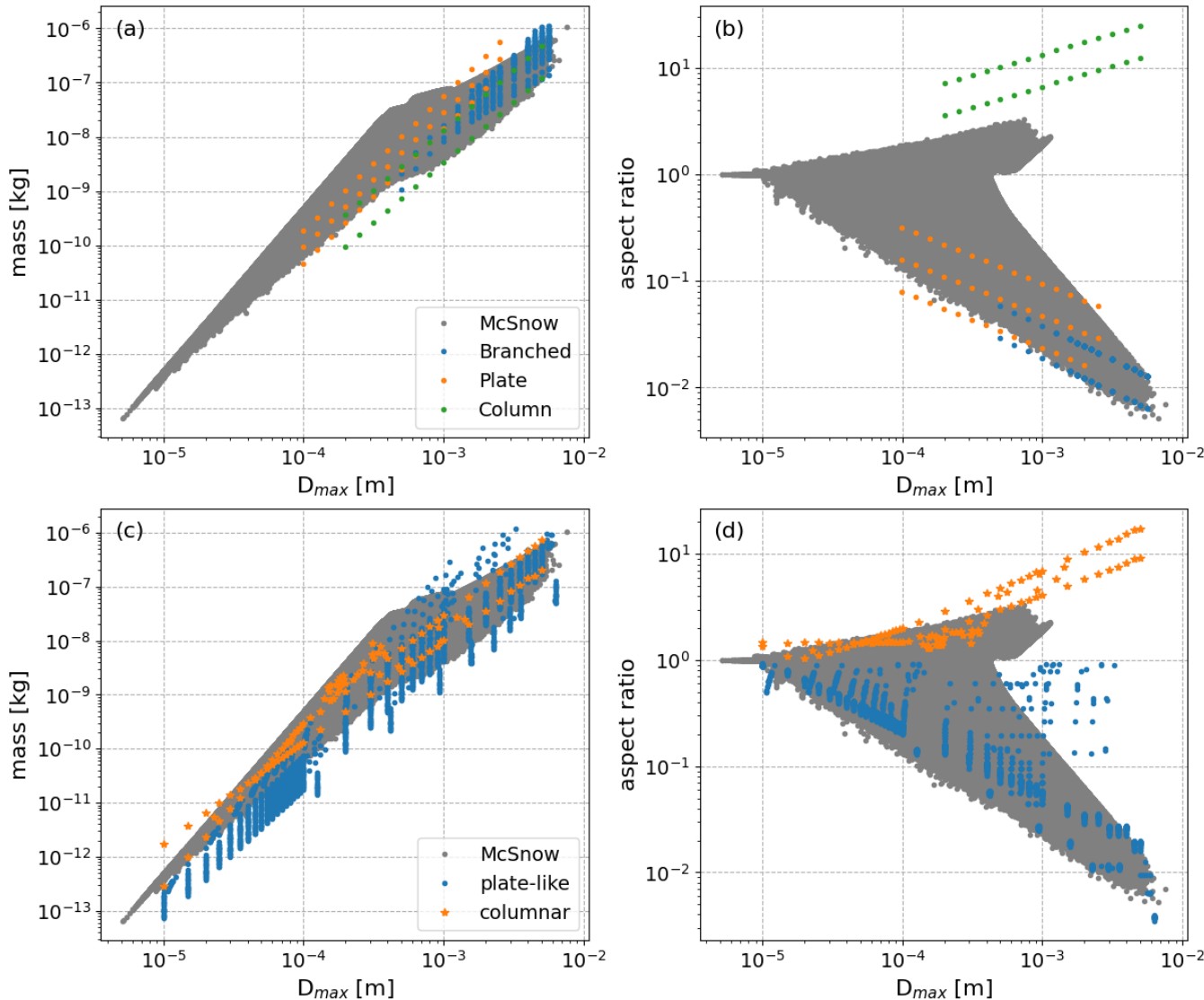

**Figure C1.** Comparing the microphysical properties of the Lu et al. (2016) database (first row) and our database (second row) with McSnow (grey points).

*Author contributions.* LvT generated the ice particles, performed the scattering calculations and provided the application examples. LvT and DO contributed to the radar forward operator. DO performed the averages over the azimuthally random oriented particles. SK acquired funding and guided the research project. All authors contributed to the text. The AI tool chatGPT was used to correct spelling and improve readability of the text.


*Competing interests.* The authors declare that they have no competing interests.

*Acknowledgements.* Contributions by L. von Terzi, S. Kneifel and D. Ori have been funded by the Deutsche Forschungsgemeinschaft
(DFG, German Research Foundation) Priority Program SPP2115 "Fusion of Radar Polarimetry and Numerical Atmospheric Modelling
Towards an Improved Understanding of Cloud and Precipitation Processes" (PROM), projects "Exploring the role of FRAGmentation of
ice particles by combining super-partIcle modelling, Laboratory studies, and polarimEtric radar observations" (FRAGILE, project number
492234709) and "Polarimetric Radar simulations with realistic Ice and Snow properties and mulTI-frequeNcy consistency Evaluation"
(PRISTINE project number 492274454). We would like to thank Dr. Axel Seifert and Dr. Christoph Siewert for their valuable help with
McSnow. We would further like to thank Dr. Fabian Jakub for helpful discussions concerning the setup of the look-up tables.

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
