# Peer review of "A Microwave Scattering Database of Oriented Ice and Snow Particles: Supporting Habit-Dependent Growth Models and Radar Applications (McRadar 1.0.0)"

_EGUsphere, 2025_

## Referee Comment (RC1)

This manuscript introduces a microwave scattering properties database of oriented ice crystal and aggregate particles focusing on radar applications. The main feature of this database is that it considers more than 3000 shapes to account for habit variations in ice microphysical schemes. The database will be very useful for radar remote sensing and data assimilation, and the manuscript is well-written. I have several suggestions as follows.

- 1. The current database covers four frequencies. Do you have plans to extend it to cover more higher frequencies and make it available to passive microwave remote sensing?
- 2. In the introduction section, only the DDA-based databases are mentioned. There are also microwave scattering property databases developed not based on the DDA method, such as: Ding, J., L. Bi, P. Yang, G. W. Kattawar, F. Weng, Q. Liu, and T. Greenwald, 2017: Single-scattering properties of ice particles in the microwave regime: temperature effect on the ice refractive index with implications in remote sensing, J. Quant. Spectrosc. Radiat. Transfer, 190, 26-27. The authors may consider also mention these studies.
- 3. In Eq. (1), the formulation of amplitude scattering matrix usually have two common forms:

$$\begin{bmatrix} S_2 & S_3 \\ S_4 & S_1 \end{bmatrix} \text{ or } \begin{bmatrix} S_{11} & S_{12} \\ S_{21} & S_{22} \end{bmatrix}$$

The form in Eq. (1)  $\begin{bmatrix} S_1 & S_4 \\ S_3 & S_2 \end{bmatrix}$  is less common. The authors may consider modify the formulation.

---

## Referee Comment (RC2)

**Review comments**

**Overall comments**

In this paper, titled "A Microwave Scattering Database of Oriented Ice and Snow Particles: Supporting Habit-Dependent Growth Models and Radar Applications (McRadar 1.0.0)," the authors presented a database of microwave scattering from ice and snow particles, which are computed using the discrete dipole approximation (DDA). The database contains scattering matrices of 2627 individual ice crystals and 450 aggregates for four microwave frequencies in C-, X-, Ka- and W-bands.

The database developed by the authors would become a new option to perform forward radar simulation with ice particles. The larger number of particle samples might be an advantage compared with past databases. However, I think the quality of the manuscript is not sufficiently high for a journal paper, and major revision would be required.

**Major comments**

- 1) Lagrangian super-particle models and habit-prediction schemes should be introduced in the introduction, probably in the fourth paragraph. These models and schemes seem to be important for the authors' motivation, according to the abstract.
- 2) In Section 2, the explanation of the methods is not sufficient. Particularly, explanations of the Reiter algorithm and the aggregation model are necessary. If a brief explanation is difficult, the authors can add an appendix for the explanation. References to the original papers would not be sufficient to explain the methods for the present study.

In the present manuscript, it is impossible to understand the meanings of the key parameters  $\alpha$ ,  $\beta$ , and  $\gamma$ , and thus the parameter values tell nothing.

It is also difficult to understand why the Reiter algorithm is not enough to cover the variety of habits because there is no explicit explanation of the detail and the limitation of the algorithm.

The aggregation model seems to be used for generating columnar crystals, while there is no explanation of the aggregation model in Subsection 2.2. What is the aggregation model and how did the authors use it?

- 3) In figures 1 and 3, it is not easy to understand the three-dimensional structure of particles only from the silhouettes. I think the authors can add the edge lines or differentiate the face color depending on the directions. Such visualization would be possible by using e.g. ParaView.
- Figure 3 is not referred to in the main text. Add the reference to figure 3 in the main text or remove figure 3.
- 4) In figure 5, the authors copied and modified an image from Wikipedia, "Euler angles" (<a href="https://en.wikipedia.org/wiki/Euler\_angles">https://en.wikipedia.org/wiki/Euler\_angles</a>). The license of this image file is CC BY 3.0. Therefore,

the authors must give appropriate credit, provide a link to the license, and indicate if changes were made.

This image in Wikipedia explains a rotation of z-x-z sequence. However, in line 153, the authors wrote that they use 3 Euler rotations "that follow the commonly used zyz-conversion (Fig. 5b)." This description is not consistent with the figure 5: The rotation sequence is different, and the image of the rotation sequence is used in figure 5a.

The notations for the three rotation angles are the same as the key parameters for Reiter algorithm. The notations for different quantities should be differentiated to avoid any confusion.

- In Section 3, there is no explanation about the discrete dipole approximation (DDA) whereas the simulation method is essentially important to consider the reliability of the simulated results. It would also be important to explain why the approximation is considered so accurate. The representation method of particle shapes and the discretization method for the dipoles should also be explained. The equations to compute  $Z_{ij}$  from  $S_i$  should also be shown to complete the explanations.
- 6) At line 193, it is assumed that snowflakes fall by aligning the longest axis of inertia horizontally, but this assumption would not be appropriate. In a recent letter by Bhowmick et al. (*Phys. Rev. Lett.*, 132, 034101,2024), existence of strong orientation fluctuations of nonspherical solid particles in the atmosphere is reported experimentally and theoretically. It is also stated that this orientation fluctuation occurs for typical atmospheric particles, and the fluctuation may be enhanced by turbulence.

Equation (7), below the sentence at line 193, does not consider the assumption, and then it seems that the authors consider the assumption to obtain Eq. (8). This relationship should be clarified.

- The scattering properties are computed by the DDA for incident microwave directions  $(\theta_0, \phi_0)$  in the particle reference frame (PRF), where the azimuth angle  $\phi_0$  is selected from  $0^o$  to  $360^o$  with interval of  $22.5^o$ . The interval  $22.5^o$  would be too coarse for dendritic ice particles to obtain sufficient statistics of scattering for arbitrary incident direction. Since the dendritic particles have six branches, only three azimuth angles are considered in the present setting. It is necessary to show the evidence that the computation is sufficiently accurate even in the present setting, if the authors think it is sufficiently accurate.
- For the case of spherical water droplet, the normalized scattering cross section  $k^2\sigma$  is given by a function of kD, where  $\sigma$  is the scattering cross section, k is the microwave wavenumber, and D is the particle diameter. This is because the electric field around the sphere is determined by the ratio of the wavelength of the electric field and the particle diameter. I imagine that the same normalization would also work for ice particles, i.e., normalized scattering properties are the same for

the same value of  $kD_{max}$  with the same shape. If it is true, the results computed by the DDA is applicable to arbitrary frequency. Why do the authors need to limit the number of frequencies for the computation?

- 9) In figure 6, the authors plotted the radar reflectivity factor, but the number density is necessary to determine the factor if I am correct. The discussion up to Subsection 3.4 is only about single scattering properties. I think the authors should explain how they assumed the number density of particles. I have the same question also to Figure 8.
- 10) The verification of the authors' DDA computation should be provided. I think that Rayleigh scattering from spherical and spheroid particles would be good examples and should be compared with theoretical solutions.
- In Section 4.1.1, the error in scattering properties caused by the regression based on the nearest-neighbor method is verified, and the authors accepted the small deviations in individual particle properties. However, there is no quantitative discussion. The authors should explain quantitatively why the error level shown in Table 1 is sufficiently small in forward radar simulations. In addition, the discussion in this section is only verification, it is not validation. The subsection title should be corrected.
- In Subsection 5.2, the authors stated that the advantage of the present database is larger number of particle samples. As the authors state, a dense database could be better than a sparse database. However, in that subsection, the authors do not discuss the advantage of their dense database. The authors should compare the databases with different numbers of samples. I think it is possible to create such database by randomly subsampling particles from the original database.
- 13) The advantage of the authors' database should be clarified in comparison with that of Lu et al. (2016). The database of Lu et al. (2016) provides scattering properties across a range of orientations and includes over 1000 ice particles, ranging from single crystals to aggregates and graupel for X-, Ku-, Ka, W-bands. The difference of the present database from that of Lu et al. seems to be the number of particles and the chois of the frequencies. Is that the advantage of the present database?
- Appendices should be referred to in the main part of the paper. The appendices should also contain main sentences that explain and discuss the figures and tables in the appendices.

**Minor comments**

- [1.45] "log" in O(nlogn) should not be italic.
- [l. 106] It would be better to write "Section 5.1".

- [1. 113] Probably, the citation should be written as "Leinonen and Moisseev (2015)".
- [1. 145 and 151] Probably, "is fixed with" would be better than "is solid with".
- [l. 156] The explanation about the angle az is not precise. The rotation with angle az does not correspond to the inverse rotation with angle  $\alpha$  since the absolute values of az and  $\alpha$  are different in general. Only the difference  $(\alpha az)$  is meaningful.
- [Eq. (2)] The variables I, Q, U, and V are not defined.
- [l. 170] The reference of Yurkin and Hoekstra (2011) is a journal paper on JQSRT, which is not the User Manual of the ADDA code. The authors should add appropriate reference information. I think the relationship of the Mueller and amplitude matrices would be found in other review papers or textbooks.
- [Subsection 4.1] There is no reference to McRadar. It seems that the authors are at least a part of the developer of McRadar, but the development of McRadar is not be the main topic of this paper. Therefore, appropriate references should be added to McRadar at least in Subsection 4.1.
- [1. 320-323] Use italic fonts for variables *c* and *a* appropriately.
- The authors must explain what the abbreviations are. For example, explanations of ZDR, DWR, and MDV are missing.
- I think the availability of the level 0 database should also be written in *Code and data availability*.

---

## Author Response (AR1)

**Reply to reviewers**

We would like to thank the reviewers for their constructive and valuable feedbacks, which we believe has greatly improved the paper. Our answers to the reviewers are written in blue.

**Reviewer 1**

This manuscript introduces a microwave scattering properties database of oriented ice crystal and aggregate particles focusing on radar applications. The main feature of this database is that it considers more than 3000 shapes to account for habit variations in ice microphysical schemes. The database will be very useful for radar remote sensing and data assimilation, and the manuscript is well-written. I have several suggestions as follows.

1. The current database covers four frequencies. Do you have plans to extend it to cover more higher frequencies and make it available to passive microwave remote sensing?
   Currently, we do not plan to include more frequencies although this would be technically possible (it is more a question of personal and computational resources). Recent developments of McSnow have rather pushed our focus to extend the range of aggregate shapes represented, aiming to better cover the natural variability of snow properties. We will publish an updated version of the database including approximately 7,000,000 aggregates. We also provide the shapefiles together with the scripts used to calculate the scattering properties to allow other scientists to extend the database to their own application of interest.

2. In the introduction section, only the DDA-based databases are mentioned. There are also microwave scattering property databases developed not based on the DDA method, such as: Ding, J., L. Bi, P. Yang, G. W. Kattawar, F. Weng, Q. Liu, and T. Greenwald, 2017: Single-scattering properties of ice particles in the microwave regime: temperature eSect on the ice refractive index with implications in remote sensing, J. Quant. Spectrosc. Radiat. Transfer, 190, 26-27. The authors may consider also mention these studies.
   We thank the reviewer for pointing out these other studies. We have intentionally only included studies based on DDA or equivalent methods, as other methods such as the T-Matrix method have been proven in recent years to misrepresent especially the polarimetric scattering properties of ice crystals and aggregates (Kneifel et al., 2020), and are thus not well suited for our application. We did not want to discuss all the difficulties of other methods that arise when looking at ice particles, thus we chose not to mention them. We also chose not to mention any studies that do not allow to forward simulate polarimetric variables (such as Ori et al. (2021), or databases based on particles with totally random orientation such as Kuo et al. (2016)), as they are again not well suited for our application.

3. In Eq. (1), the formulation of amplitude scattering matrix usually have two common forms is less common. The authors may consider modify the formulation.
   Yes, the reviewer is correct, there are other common forms, however, we have made the convention here consistent with the one used by ADDA, therefore we would prefer to keep it as is, so that it is easier for the users of the database and for the further extension of the database by other scientists who could leverage on the codes and scripts we provided.

**Reviewer 2**

Overall comments In this paper, titled "A Microwave Scattering Database of Oriented Ice and Snow Particles: Supporting Habit-Dependent Growth Models and Radar Applications (McRadar 1.0.0)," the authors presented a database of microwave scattering from ice and snow particles, which are computed using the discrete dipole approximation (DDA). The database contains scattering matrices of 2627 individual ice crystals and 450 aggregates for four microwave frequencies in C-, X-, Ka- and W-bands. The database developed by the authors would become a new option to perform forward radar simulation with ice particles. The larger number of particle samples might be an advantage compared with past databases. However, I think the quality of the manuscript is not sufficiently high for a journal paper, and major revision would be required.

**Major comments**

**Comment 1**

Lagrangian super-particle models and habit-prediction schemes should be introduced in the introduction, probably in the fourth paragraph. These models and schemes seem to be important for the authors' motivation, according to the abstract.

Indeed we use a Lagrangian Super-particle model (McSnow) in this paper but it is not the only model that applies habit prediction schemes. Other studies have also used habit prediction schemes in bin-models (Hashino and Tripoli, 2007) or even bulk models (Jensen et al., 2017). We do not want to give the (wrong) impression that habit prediction is exclusively a method used in Lagrangian super-particle models. In fact, our database might be relevant to any other model that employs complex ice microphysics. We have changed the paragraph in the introduction to include more details about habit prediction and Lagrangian particle models, while also including the other models that use habit prediction schemes.

**Comment 2**

In Section 2, the explanation of the methods is not sufficient. Particularly, explanations of the Reiter algorithm and the aggregation model are necessary. If a brief explanation is difficult, the authors can add an appendix for the explanation. References to the original papers would not be sufficient to explain the methods for the present study. In the present manuscript, it is impossible to understand the meanings of the key parameters $\alpha$, $\beta$, and $\gamma$, and thus the parameter values tell nothing. It is also difficult to understand why the Reiter algorithm is not enough to cover the variety of habits because there is no explicit explanation of the detail and the limitation of the algorithm. The aggregation model seems to be used for generating columnar crystals, while there is no explanation of the aggregation model in Subsection 2.2. What is the aggregation model and how did the authors use it?

We have added a detailed description of the Reiter algorithm to section 2.1, and a brief explanation of the aggregation model to section 2. We believe that a more in-depth description of the aggregation model is out of the scope of this paper because we did not produce any new aggregates, they have been generated and described in Ori et al. (2021); Karrer et al. (2020). Also, the aggregation model itself has been used and described in various previous studies (Leinonen and Moisseev, 2015; Leinonen and Szyrmer, 2015; Leinonen et al., 2018; Ori et al., 2021; Maherndl et al., 2023, 2025; Leinonen et al., 2021; Köbschall et al., 2023; Seifert et al., 2019), and we think it can therefore be considered to be sufficiently well established. We have changed Section 2 and Section 2.1 to include the description of the Reiter algorithm and discretization of the particles.

**Comment 3**

In figures 1 and 3, it is not easy to understand the three-dimensional structure of particles only from the silhouettes. I think the authors can add the edge lines or differentiate the face colour depending on the directions. Such visualization would be possible by using e.g. ParaView. Figure 3 is not referred to in the main text. Add the reference to figure 3 in the main text or remove figure 3.

Thanks for the suggestion. Finding a way to best show the complex shapes of snow crystals and aggregates is difficult. Paraview and its capability to project light and cast shadows of 3D object is indeed a nice feature, but works best if the 3D objects represented are in a limited number of individual solid shapes. Our shapes are described by the 3D locations of hundreds of thousands or millions of individual, tiny, cubic

voxels. The overall shape of individual monomers is lost in the process of voxelization. Other scattering databases such as ARTS (see Fig1 and 2 of https://essd.copernicus.org/articles/10/1301/2018/essd-10-1301-2018.pdf) used Paraview or similar software to show the particles shape including the details of the monomers, but this approach, although visually pleasing, does not present the actual particle shape that is used for the scattering simulations (see blue images in those ARTS figures). Moreover, for fractal shapes like dendrites they also inevitably had to fallback to a per-voxel visualization (see grey images in the ARTS figures) which gives equivalent image quality to our visualization technique. We tried to plot the voxel edges as well, but the result was rather unsatisfactory due to the immense amount of overlapping colors. We preferred than to keep the current version of the figure as our best attempt.

**Comment 4**

In figure 5, the authors copied and modified an image from Wikipedia, "Euler angles" (`https://en.wikipedia.org/wiki/Euler_angles`). The license of this image file is CC BY 3.0. Therefore, the authors must give appropriate credit, provide a link to the license, and indicate if changes were made. This image in Wikipedia explains a rotation of z-x-z sequence. However, in line 153, the authors wrote that they use 3 Euler rotations "that follow the commonly used zyz-conversion (Fig. 5b)." This description is not consistent with the figure 5: The rotation sequence is different, and the image of the rotation sequence is used in figure 5a. The notations for the three rotation angles are the same as the key parameters for Reiter algorithm. The notations for different quantities should be differentiated to avoid any confusion.

Thanks for spotting this we substituted the portion of the figure with the appropriate image that as been adapted from the ADDA User Manual. Authorship has been mentioned in the Figure caption and we also added a Copyright statement.

**Comment 5**

In Section 3, there is no explanation about the discrete dipole approximation (DDA) whereas the simulation method is essentially important to consider the reliability of the simulated results. It would also be important to explain why the approximation is considered so accurate. The representation method of particle shapes and the discretization method for the dipoles should also be explained. The equations to compute $Z_{ji}$ from $S_i$ should also be shown to complete the explanations.

We have added section 3.1 (see answer to comment 10) which explains the DDA. We have added a more thorough description of how the particles are obtained and discretized in Section 2 (see answer to comment 2). Providing all equations to convert $Z_{ji}$ from $S_i$ is not feasible in the context of this paper, as it is a large set of equations. For convenience we have added the conversion of the four elements relevant for radar applications to the manuscript (equations 3 to 6). If the reader is interested in the full conversion, all code, including the conversion from $S_i$ to $Z_{ji}$ is published on github.

**Comment 6**

At line 193, it is assumed that snowflakes fall by aligning the longest axis of inertia horizontally, but this assumption would not be appropriate. In a recent letter by Bhowmick et al. (Phys. Rev. Lett., 132, 034101,2024), existence of strong orientation fluctuations of nonspherical solid particles in the atmosphere is reported experimentally and theoretically. It is also stated that this orientation fluctuation occurs for typical atmospheric particles, and the fluctuation may be enhanced by turbulence. Equation (7), below the sentence at line 193, does not consider the assumption, and then it seems that the authors consider the assumption to obtain Eq. (8). This relationship should be clarified.

We agree on the fact that snowflakes do not always fall with their longest axis aligned horizontally. However we think that we we sufficiently mentioned this aspect in the first two paragraphs of section 3.3 (now section 3.4). As suggested by the reviewer, we mention the presence of strong fluctuation and its enhancement due to turbulence. We also believe that Eq.7 (now Eq.11) does precisely the job of considering the assumption that distributions of orientations exist and we provide the general method to obtain orientation averaged scattering properties. Eq.8 (now Eq.12) performs a simplification by imposing that there is no preferential direction on the horizontal plane, hence the distributions of $\alpha$ and $\gamma$ are uniform and equivalent to $1/2\pi$, thus these 2 variables can be readily integrated over their assumed distribution, reducing the dimensionality of the problem of 2 free parameters. The distribution of $\beta$ and the radar elevation angle remain free parameters of the ARO scheme (note also that the radar azimuth was never taken into account because it is equivalent to the $\alpha$ angle).

We have moved the definition of the "default orientation" of the snowflakes to better clarify that its

purpose is only to define the axis around which the Euler rotations of the particles are performed and we tried to clarify our meaning in the revised text

**Comment 7**

The scattering properties are computed by the DDA for incident microwave directions ($\theta_0$, $\phi_0$) in the particle reference frame (PRF), where the azimuth angle $\phi_0$ is selected from 0° to 360° with interval of 22.5°. The interval 22.5° would be too coarse for dendritic ice particles to obtain sufficient statistics of scattering for arbitrary incident direction. Since the dendritic particles have six branches, only three azimuth angles are considered in the present setting. It is necessary to show the evidence that the computation is sufficiently accurate even in the present setting, if the authors think it is sufficiently accurate.

Because we consider azimuthally randomly oriented particles, only a limited number of azimuth angles is required, as the scattering properties are averaged over all azimuths. To verify this, we performed additional calculations for a randomly selected dendrite using twice the original number of azimuth angles (32 angles with 11.25° spacing). We also subsampled the dataset to half (8 angles, 45° spacing) and one quarter (4 angles, 90° spacing) of the original azimuthal resolution. We then examined the dependence of the backscattering cross section, ZDR, and KDP on the angle $\beta$ (see Figure 1).

The results show that even substantially fewer azimuth angles would have been sufficient to obtain stable azimuthally averaged scattering properties. Minor deviations appear when only 8 azimuth angles are used, particularly for $\beta < 30°$, and these deviations become more pronounced when the number is reduced to 4. Nonetheless, the overall impact on the azimuthally averaged quantities remains small.

[Figure]

Figure 1: Backscattering cross section (first panel), ZDR (second panel) and KDP (third panel) in dependency of $\beta$ assuming azimuthally random orientations.

**Comment 8**

For the case of spherical water droplet, the normalized scattering cross section $k^2\sigma$ is given by a function of $kD$, where $\sigma$ is the scattering cross section, $k$ is the microwave wavenumber, and $D$ is the particle diameter. This is because the electric field around the sphere is determined by the ratio of the wavelength of the electric field and the particle diameter. I imagine that the same normalization would also work for ice particles, i.e., normalized scattering properties are the same for the same value of $kD_{max}$ with the same shape. If it is true, the results computed by the DDA is applicable to arbitrary frequency. Why do the authors need to limit the number of frequencies for the computation?

This is an interesting idea! What you are proposing is essentially that the scattering only varies with the size parameter X. However, this would only be true if the refractive index does not change between wavelengths. Although the refractive index does not change by large amounts for ice in the microwave region it cannot be considered constant (especially the imaginary part). Moreover, it is definitely more convenient for a user to queue the database using physical quantities such as particle size and mass rather than using dimensionless parameters. In fact, scaling the dimensions would make the range of represented sizes also too depend on frequency which would make the use of the database quite inconvenient especially for multi-frequency radar applications. Finally, the biggest issue comes from the fact that snowflakes do not retain the same overall shape at different sizes. Large snowflakes whose growth is driven by aggregation and individual dendritic crystals are known to exhibit a fractal behaviour (Westbrook et al., 2004); even simpler pristine shapes such as solid plates, columns and needle are found to have a size-dependent aspect-ratio (Jensen et al., 2017) which makes their mass to scale with size with a power-law with an exponent smaller than 3. The frequency-scaling approach is physically reasonable only on scattering targets that retain their overall shape at any size (which is the case for spherical water drops)

**Comment 9**

In figure 6, the authors plotted the radar reflectivity factor, but the number density is necessary to determine the factor if I am correct. The discussion up to Subsection 3.4 is only about single scattering properties. I think the authors should explain how they assumed the number density of particles. I have the same question also to Figure 8.

The reviewer is technically correct. We have changed the figures andshow the backscattering cross sections multiplied by $wl^4/(\pi^5 * |K|^2)$. A note was also added to the caption.

**Comment 10**

The verification of the authors' DDA computation should be provided. I think that Rayleigh scattering from spherical and spheroid particles would be good examples and should be compared with theoretical solutions.

The DDA is widely regarded as an accurate method for computing electromagnetic scattering, provided that the dipole spacing $d$ is sufficiently small. A commonly applied accuracy criterion is $|m|kd < 0.05$ (Tyynelä et al., 2009; Leinonen and Moisseev, 2015; Zubko et al., 2010), where $m$ is the refractive index and $k$ the wavenumber. At X-band (9.4 GHz), this corresponds to a maximum dipole spacing of approximately 0.14 mm. The largest spacing used in our study is 0.01 mm, and the criterion is therefore comfortably satisfied.

A second source of uncertainty concerns the discretization of particle geometry, which is particularly critical for strongly asymmetric shapes, such as thin plates, whose thickness may span only a single layer of dipoles. To address this, we adopted a variable dipole spacing that depends on the particle maximum dimension. To assess whether this spacing is sufficiently small, we selected two random dendritic particles from our database and reduced the dipole spacing by factors of two and four. The resulting changes in the computed scattering properties were only 0.12–0.2% (see Fig. 4), depending on polarization and wavelength, indicating that the discretization is more than adequate.

Previous studies have compared DDA calculations with those from the T-matrix method, which provides exact solutions for homogeneous particles (Tyynelä et al., 2009). These comparisons show that DDA backscattering cross sections for pure-ice particles typically differ from T-matrix results by only 1.7–2.6%. However, similar validation cannot be performed for our dataset, as the complex, highly irregular particle shapes considered here cannot be represented within the T-matrix framework. Large discrepancies would be expected particularly for low-density structures such as dendrites.

We emphasize that the dominant source of uncertainty in this study does not arise from the DDA computations themselves but rather from the choice of particle shapes. As illustrated in Fig. 6 in the

manuscript, the scattering properties vary substantially across different particle geometries. Because the true shapes of atmospheric ice particles are unknown, this introduces a significant yet currently unquantifiable uncertainty. The lack of suitable in-situ observations prevents a systematic assessment of this error. Section 3.1 of the revised manuscript now provides additional discussion of the DDA methodology and its potential limitations. We have added the subsection 3.1 to the manuscript which describes the DDA as well as its uncertainties.

**Comment 11**

In Section 4.1.1, the error in scattering properties caused by the regression based on the nearest-neighbour method is verified, and the authors accepted the small deviations in individual particle properties. However, there is no quantitative discussion. The authors should explain quantitatively why the error level shown in Table 1 is sufficiently small in forward radar simulations. In addition, the discussion in this section is only verification, it is not validation. The subsection title should be corrected.

In our opinion, Figure 7 in the manuscript illustrates well that the selected values scatter randomly around the expected values, and no clear bias is visible. Because the reconstruction errors are random and unbiased, they do not accumulate in forward simulations where radar variables (Ze, ZDR, KDP, or others) are obtained by considering over hundreds to thousands of particles. We have added the last sentence to section 4.1.1 to better explain our reasoning. We have also changed the title of the subsection to "Evaluation of the Nearest Neighbour Lookup".

**Comment 12**

In Subsection 5.2, the authors stated that the advantage of the present database is larger number of particle samples. As the authors state, a dense database could be better than a sparse database. However, in that subsection, the authors do not discuss the advantage of their dense database. The authors should compare the databases with different numbers of samples. I think it is possible to create such database by randomly subsampling particles from the original database.

We thank the reviewer for this interesting idea! We performed an experiment where we randomly downsampled our database to contain 1/3 of the original LUT, and performed the forward simulations of the McSnow simulation formerly presented in Figure 9. We did this 100 times in total. We need to account for the smaller number of particles in the downsampled LUT. Assuming that the LUT is populated evenly, the smaller number of particles can be considered by adjusting the number of particles used to calculate the scattering properties of each McSnow particle. 10 was chosen for the full LUT, 3 was chosen for the downsampled LUT. The results are shown in Figure 2. This experiment nicely shows the advantage of our database: the large number of particles, which allows the particle properties space to be filled much more densely, therefore 1) allowing to select the particle which fits best to the McSnow simulation and 2) reducing the statistical noise by being able to average the scattering properties of multiple particles, which might in turn reduce the uncertainty introduced by not knowing what the exact shape of ice particles in the atmosphere is. As expected, the reduction of the number of particles in the LUT increases the random fluctuations ("noise"). The simulations vary around the original simulation, where the entire database was considered. This variability increases if e.g. only 1/4th of the original particles are chosen for the LUT. We modified our original Section 5.2 to contain the results of this experiment.

**Comment 13**

The advantage of the authors' database should be clarified in comparison with that of Lu et al. (2016). The database of Lu et al. (2016) provides scattering properties across a range of orientations and includes over 1000 ice particles, ranging from single crystals to aggregates and graupel for X-, Ku-, Ka, W-bands. The difference of the present database from that of Lu et al. seems to be the number of particles and the chois of the frequencies. Is that the advantage of the present database?

The particles used in the Lu et al. (2016) database have only a limited amount of variability especially in their aspect ratios. In Figure 3 we have plotted the aspect ratios and masses found in the database for branched crystals, plates and columns. It is evident that the amount of microphysical properties in our database is much larger, especially the variability in aspect ratio, which is needed in order to forward simulate the output of models using a habit prediction for ice crystals. We have added a sentence to the new Section 5.2 to clarify this.

[Figure]

Figure 2: Variability of scattering properties if only 1/3 of particles from the database is used to forward simulate the McSnow simulation. Shown are Ze (a), MDV (b), $DWR_{KaW}$ (c), ZDR (d), KDP (e) and sZDRmax (f). The black line indicates the original simulation, where the entire database was used.

**Minor comments**

- l. 45: "log" in O(nlogn) should not be italic. Fixed

- l. 106: It would be better to write "Section 5.1". Fixed

- l. 113: Probably, the citation should be written as "Leinonen and Moisseev (2015)". Fixed

- l. 145 and 151 Probably, "is fixed with" would be better than "is solid with". Fixed

- l. 156 The explanation about the angle az is not precise. The rotation with angle az does not correspond to the inverse rotation with angle $\alpha$ since the absolute values of az and $\alpha$ are different in general. Only the difference $(\alpha-$ az$)$ is meaningful. Right, fixed in the text

- Eq. (2) The variables I, Q, U, and V are not defined. we have added a brief description of the elements of the Stokes vector below equation 2.

- l. 170 The reference of Yurkin and Hoekstra (2011) is a journal paper on JQSRT, which is not the User Manual of the ADDA code. The authors should add appropriate reference information. I think the relationship of the Mueller and amplitude matrices would be found in other review papers or textbooks. We have changed this paragraph, no reference to Yurkin and Hoekstra is needed anymore. We describe how the Mueller and Amplitude matrices are related briefly. See also answer to comment 5.

- Subsection 4.1 There is no reference to McRadar. It seems that the authors are at least a part of the developer of McRadar, but the development of McRadar is not be the main topic of this paper. Therefore, appropriate references should be added to McRadar at least in Subsection 4.1. We are the developers of McRadar, and McRadar has thus far not been published anywhere.

- l. 320-323 Use italic fonts for variables c and a appropriately. Fixed

[Figure]

Figure 3: Ice particle mass and aspect ratio as a function of the maximum dimension ($D_{max}$) for the ice crystals in the Lu et al. (2016) database (a,b) and from our database (c,d). The particle properties obtained from a McSnow simulation are shown in grey.

- The authors must explain what the abbreviations are. For example, explanations of ZDR, DWR, and MDV are missing.
  We introduced the abbreviations of ZDR, DWR and MDV in the manuscript

- I think the availability of the level 0 database should also be written in Code and data availability.
  We have published the lv0 dataset here: von Terzi (2025)

**Dr. Maxim Yurkin**

I find this paper very interesting and I am fascinated by the amount of DDA simulations performed by the authors. I have a few technical comments about these DDA simulations (mostly about their description in the paper):

1. Eq.(2) is probably taken from ADDA manual. However, in the case of the atmosphere, one can probably always replace ksca by just k, which is always real. Then exponent will disappear and absolute value around k is redundant. Yes, you are correct, we have edited that in the manuscript.

2. Concerning the name of the ADDA code, the official guideline is not to deabbreviate it - see `https://github.com/adda-team/adda/wiki/FAQ#what-is-the-official-name-of-the-code-what-does-a-stands-`. In other words, the standard naming is just ADDA. Fixed.

3. In Sec. 3.3 there is a rather extensive description of assumptions concerning the orientation probability distribution. The authors may additionally note, as a first step, that general distribution $f(\alpha, \beta, \gamma)$ is assumed to be factorizable into three parts (it is not always possible). I guess, such factorization can be considered as a part of ARO assumption that is introduced further in the manuscript text. This is an interesting point which we believe not many people even thought about. We mention this aspectnow in section 3.3.

4. I could not find a description of the ADDA parameters used for simulations. It would be great to specify them for the overall reproducibility, including the ADDA version, DDA formulation, and discretization. Mentioning that some parameters are set to default values will also be fine.
The authors mention that there exist level 0 with all the log files (which may answer the above question), but it seems that only levels 1 are available online, or not?
We have decided to also publish the Level 0 dataset, due to its size it is not stored on Zenodo but on the LMU server: von Terzi (2025)
Some information can be reduced from the scripts to run ADDA simulations (thanks to authors for sharing them), for instance, this one - `https://github.com/lterzi/DDA_database_gmd/blob/master/calculate_dda/DDA_aggregates_plates/print_adda_command_elevation.py` . It contains the line:

opt='-pol ldr -int fcd -iter qmr2 -scat_matr both -dir '

This is surprising for me, since it combines LDR with FCD. Overall, I do not expect the results to depend significantly on the choice of polarizability (i.e., the results for '-pol fcd -int fcd ...' will probably be close). However, I am wandering, why the authors decided to use such a combination. If there is some explanation (rationale), it would be great to include it in the paper.
We admit that we did not pay much attention to this. As Dr. Yurkin correctly notes, it is rather odd to mix the ldr polarization with fcd as interaction term. This is a result of having experimented in the past with different combinations of ADDA settings in search of the best possible setup for our applications. After finding fcd-fcd to perform best for scattering targets composed of multiple media with high contrast in dielectric properties (i.e. melting snow (Ori and Kneifel, 2018)) we just fall back to ldr as polarization formulation since it gives a considerable performance boost in case of dielectrically tenuous targets such as ice in the microwave. We give a short discussion of the setup in a new subsection dedicated to DDA

5. Whatever is the used DDA formulation and other parameters, the really important measure is the uncertainty of the DDA simulations (expected errors). Is there any estimate on that? I see '-jagged ...' in the above-mentioned script, which suggests that the authors did some accuracy studies. I understand that, after all the averaging, the DDA uncertainties will most probably be negligible in comparison with other error sources. Still, just assuming that DDA results are exact seems to be not robust.
Yes, you are correct, azimuthal averaging should mitigate uncertainties, but the most prominent source of uncertainty is our primary lack of knowledge of what is the actual shape of a snowflake. Considering the fact that weather radars sample millions of ice particles at the same time it is already a big step forward even attempting to capture that variability with a larger dataset. For our database, we have adopted a variable dipole spacing depending on the maximum Dimension of the ice crystals, to ensure that the small details of the particles are captured also for very small

particles. This makes it difficult to decide which particles to select to test the convergence of the discretization errors. However, we have selected two dendrites (one small and one large) with a large number of details (such as fine branches) and performed a sensitivity study by reducing the dipole spacing by a factor of 2 and 4 and analysed the differences to our original dipole spacing for all calculated azimuth angles at 30 and 90° elevation. One example plot is provided in Figure 4. The total error (difference between 2 (4) times smaller dipole spacing and the original dipole spacing) ranges between 0.10 and 0.2% depending on the wavelength, polarization, azimuth and elevation angle, indicating that the discretization is sufficiently fine.

**References**

S. Kneifel, J. Leinonen, J. Tyynelä, D. Ori, and A. Battaglia. Scattering of hydrometeors. In *Satellite precipitation measurement*, pages 249–276. Springer, 2020. doi: 10.1007/978-3-030-24568-9_15.

D. Ori, L. von Terzi, M. Karrer, and K. Kneifel. snowscatt 1.0: Consistent model of microphysical and scattering properties of rimed and unrimed snowflakes based on the self-similar rayleigh-gans approximation. *Geoscientific Model Development*, 14:1511–1531, 3 2021. ISSN 19919603. doi: 10.5194/gmd-14-1511-2021.

Kwo-Sen Kuo, William S Olson, Benjamin T Johnson, Mircea Grecu, Lin Tian, Thomas L Clune, Bruce H van Aartsen, Andrew J Heymsfield, Liang Liao, and Robert Meneghini. The microwave radiative properties of falling snow derived from nonspherical ice particle models. part i: An extensive database of simulated pristine crystals and aggregate particles, and their scattering properties. *Journal of Applied Meteorology and Climatology*, 55(3):691–708, 2016. doi: 10.1175/JAMC-D-15-0130.1.

T Hashino and GJ Tripoli. The spectral ice habit prediction system (ships). part i: Model description and simulation of the vapor deposition process. *Journal of the Atmospheric Sciences*, 64(7):2210–2237, 2007. doi: 10.1175/JAS3963.1.

Anders A Jensen, Jerry Y Harrington, Hugh Morrison, and Jason A Milbrandt. Predicting ice shape evolution in a bulk microphysics model. *Journal of the Atmospheric Sciences*, 74(6):2081–2104, 2017. doi: 10.1175/JAS-D-16-0350.1.

M Karrer, A Seifert, C Siewert, D Ori, A von Lerber, and S Kneifel. Ice particle properties inferred from aggregation modelling. *Journal of Advances in Modeling Earth Systems*, 12(8):e2020MS002066, 2020. doi: 10.1029/2020MS002066.

J. Leinonen and D. Moisseev. What do triple-frequency radar signatures reveal about aggregate snowflakes? *Journal of Geophysical Research: Atmospheres*, 120(1):229–239, 2015. doi: 10.1002/2014JD022072.

J. Leinonen and W. Szyrmer. Radar signatures of snowflake riming: a modeling study. *Earth and Space Science*, page 2015EA000102, 2015. doi: 10.1002/2015EA000102.

J. Leinonen, S. Kneifel, and R. J. Hogan. Evaluation of the rayleigh–gans approximation for microwave scattering by rimed snowflakes. *Quarterly Journal of the Royal Meteorological Society*, 144:77–88, 2018. doi: 10.1002/qj.3093.

Nina Maherndl, Maximilian Maahn, Frederic Tridon, Jussi Leinonen, Davide Ori, and Stefan Kneifel. A riming-dependent parameterization of scattering by snowflakes using the self-similar rayleigh–gans approximation. *Quarterly Journal of the Royal Meteorological Society*, 149(757):3562–3581, 2023. doi: 10.1002/qj.4573.

Nina Maherndl, Alessandro Battaglia, Anton Kötsche, and Maximilian Maahn. Riming-dependent snowfall rate and ice water content retrievals for w-band cloud radar. *Atmospheric Measurement Techniques*, 18(14):3287–3304, 2025. doi: 10.5194/amt-18-3287-2025.

Jussi Leinonen, Jacopo Grazioli, and Alexis Berne. Reconstruction of the mass and geometry of snowfall particles from multi-angle snowflake camera (masc) images. *Atmospheric Measurement Techniques*, 14 (10):6851–6866, 2021. doi: 10.5194/amt-14-6851-2021.

[Figure]

Figure 4: Backscattering cross sections at horizontal polarization and 0° elevation of one particle for different dipole spacing/grid resolutions (blue: original dipole spacing of $2.5 \cdot 10^{-6}$m, orange: half of original dipole spacing, green: quarter of the original dipole spacing), for all calculated azimuth angles. The maximum absolute errors between the quarter/half resolution and original resolution for each radar wavelength at elevation=0° are provided in each panel.

Kilian Köbschall, Jan Breitenbach, Ilia V Roisman, Cameron Tropea, and Jeanette Hussong. Geometric descriptors for the prediction of snowflake drag. *Experiments in Fluids*, 64(1):4, 2023. doi: 10.1007/s00348-022-03539-x.

A. Seifert, J. Leinonen, C. Siewert, and S. Kneifel. The geometry of rimed aggregate snowflakes: A modeling study. *Journal of Advances in Modeling Earth Systems*, 11(3):712–731, 2019. doi: 10.1029/2018MS001519.

Christopher David Westbrook, RC Ball, PR Field, and Andrew J Heymsfield. Universality in snowflake aggregation. *Geophysical research letters*, 31(15), 2004. doi: 10.1029/2004GL020363.

Jani Tyynelä, Timo Nousiainen, Sabine Göke, and Karri Muinonen. Modeling c-band single scattering properties of hydrometeors using discrete-dipole approximation and t-matrix method. *Journal of Quantitative Spectroscopy and Radiative Transfer*, 110(14-16):1654–1664, 2009.

E. Zubko, D. Petrov, Y. Grynko, Y. Shkuratov, H. Okamoto, K. Muinonen, T. Nousiainen, H. Kimura, T. Yamamoto, and G. Videen. Validity criteria of the discrete dipole approximation. *Applied optics*, 49(8):1267–1279, 2010. doi: 10.1364/AO.49.001267.

Y. Lu, Z. Jiang, K. Aydin, J. Verlinde, E. E. Clothiaux, and G. Botta. A polarimetric scattering database for non-spherical ice particles at microwave wavelengths. *Atmospheric Measurement Techniques*, 9(10):5119–5134, 2016. doi: 10.5194/amt-9-5119-2016. URL `https://amt.copernicus.org/articles/9/5119/2016/`.

Leonie von Terzi. Level 0 data for: A microwave scattering database of oriented ice and snow particles: Supporting habit-dependent growth models and radar applications, September 2025. URL `https://doi.org/10.57970/cgtek-fen77`.

Davide Ori and Stefan Kneifel. Assessing the uncertainties of the discrete dipole approximation in case of melting ice particles. *Journal of Quantitative Spectroscopy and Radiative Transfer*, 217:396–406, 2018. ISSN 0022-4073. doi: https://doi.org/10.1016/j.jqsrt.2018.06.017. URL `https://www.sciencedirect.com/science/article/pii/S0022407318302942`.

---

## Referee Report (RR1)

Review comments

Overall comments

This is the second review report on the paper, titled "A Microwave Scattering Database of Oriented Ice and Snow Particles: Supporting Habit-Dependent Growth Models and Radar Applications (McRadar 1.0.0)". Thanks to the authors' response to my comments and their effort to revise the manuscript, most of my major concerns have been solved. The quality of the revised manuscript has been improved, and the advantage of the present database compared with past database has become clearer. The revised manuscript would be acceptable for publication after fixing remaining minor points. In the following, I write my response to the authors' replies. I omit my first review comments and the authors' replies, leaving the number for each major comment.

Major comments
**Comment 1**
I'm satisfied with the authors' revision in the introduction.

**Comment 2**
I'm mostly satisfied with the authors' revision and the clarification that the authors used the Reiter algorithm implemented in the aggregation model software to generate ice crystal shapes.

**Comment 3**
I understand the authors' decision to keep the present style of visualization.

**Comment 4**
Thanks to the authors' response, I understand that the figure was copied from a different source and the description of zyz-conversion is correct. However, two minor points remain: the zyz-conversion is shown in Fig. 5a; the notation of the three rotation angles ($\alpha$, $\beta$, $\gamma$) should be different from the parameters for Reiter algorithm.

**Comment 5**
I confirmed that explanations about the DDA have been added in Section 3.1, and the references for the simulation setup have been specified in the last paragraph of Section 3.1. The explanation about the accuracy of the DDA has also been added.
The discretization of particles is explained in the last part of Section 2.1.
I understand the reason why the authors avoid showing the full set of equations. The reference to Mishchenko (2002) would help the readers who are interested in it.

**Comment 6**

According to the authors' response, the authors did not assume that snowflakes fall by aligning the longest axis of inertia horizontally whereas they defined the Euler rotation angles with respect to the default orientation where the longest axis of inertia horizontally. In that sense, the sentence "we assumed that snowflakes fall by default by aligning their longest axis of inertia horizontally." Is quite confusing even in the revised manuscript, though it was good that the authors moved this sentence to the second paragraph. Since the authors use the word "fall" in this sentence, it seems that the authors assumed the physical condition about the particle settling. The orientation is not relevant to particle settling but only a relationship between two coordinates. The term "default fall configuration" should be written as "default orientation", and the description "we assumed that snowflakes fall by default by" should be just "the default orientation is defined as". Furthermore, the default orientation is not an assumption, it is a definition.

**Comment 7**

I'm satisfied with the authors' response since the accuracy is confirmed by numerical data. It would be good to explain this fact in the manuscript.

**Comment 8**

I'm satisfied with the authors' response, which clearly answered my question.

**Comment 9**

I'm satisfied with the authors' response and revision.

**Comment 10**

I'm satisfied with the authors' response and revision. It is good that the authors reported the numerical convergence changing the resolution.

**Comment 11**

The authors' response and revision are reasonable. The revised subsection title is fair.

**Comment 12**

I confirmed that the authors largely revised the discussion in Section 5.2. I think that the authors succeeded in showing the advantage of their database.

**Comment 13**

I'm satisfied with the authors' response. It is good that the authors added the new figure for the comparison of the dataset in parameter space in Appendix C.

**Comment 14**

In my first review report, my 14th comment was "Appendices should be referred to in the main part of the paper. The appendices should also contain main sentences that explain and discuss the figures and tables in the appendices."

There was no response to this comment from the authors, but I confirmed that the authors revised the manuscript partially responding to it. The remaining point is that the authors should refer to Appendix B in the main part of the manuscript.

Minor comments

The authors responded to all the minor comments in my first review report. I'm satisfied with those responses. Now I have some additional comments concerning revised sentences.

- [l. 61] Close the parentheses as "(see Section 5.1)".
- [l. 218] The Stokes vector: "S" should be capitalized.
- [l. 491] There is a strange symbol before 0.1. Is it "between $-0.1$ and $1.2°$ km$^{-1}$"?

---

## Author Response (AR2)

**Answers to Review Comments**

We would like to thank the reviewer for his answers to our review. We have changed the manuscript as detailed below.

**Comment 4**: we have changed the naming convention of the Euler angles to $\hat{\alpha}, \hat{\beta}, \hat{\gamma}$

**Comment 6**: we have made the suggested changes to the text (changed "we assumed that snowflakes fall by default by" to "the default orientation is defined as")

**Comment 7**: We have added the following two sentences to Section 3.3: Such a coarse azimuthal sampling is sufficient because the scattering properties are later averaged over azimuth (see Section~\ref{sec:AROavg}). Numerically, an azimuthal resolution as coarse as 45° yields identical azimuthally averaged scattering properties.

**Comment 14**: we have added a reference to Appendix B to Section 4.1.1: Appendix B shows the corresponding tables for the radar relevant Mueller and Amplitude matrix entries.

**Minor comments:**
- [l. 61] Close the parentheses as "(see Section 5.1)":
  Done
- [l. 218] The Stokes vector: "S" should be capitalized.
  Done
- [l. 491] There is a strange symbol before 0.1. Is it "between −0.1 and 1.2° km-1"?
  Yes it is -0.1, we fixed it.

---

## Author Response (AR3)

**Addressing the technical corrections provided by the Editor:**

- page 2 / line 24 - use a persistent URL: https://web.archive.org/web/20220210055742/https://raw.githubusercontent.com/adda-team/adda/v1.4.0/doc/manual.pdf and consider including this work in References (with a reference from Fig. 5 caption) Done

- Fig. 1: top panels have unreadable cluttered vertical axis tick labels. Fixed

- Table 1 - first column: "bck" subscripts, ZDR and KDP should be typeset in upright font. Fixed

- Fig. 5 contains raster elements, consider replacing these with newly constructed vector graphics (or at least clean up grey leftover pixels around x', y' and z' axis labels). We have cleaned up the grey leftover pixels

- add punctuation to the equations (i.e. full stops and commas in \text{} mode). Done

- the contents of the https://github.com/lterzi/DDA_database_gmd repository need to be persistently archived (e.g., Zenodo) and annotated accordingly in the Code & Data availability section. We have adapted the codes of DDA_database_gmd with the changes made during the review process and published the changes as a new version on zenodo. We have updated the code and data availability accordingly.

- the DOI in footnote 1 is invalid (10.57970/cgtek-fen7). Fixed

- page 5 / line 153: do not use italic font for "grid" or "res". Fixed

- captions of Fig. 1, Fig. 2, Fig. 4, Fig. 8 and Fig. 9, and 4 occurrences in text on pages 19 and 27: use upright font for the "max" subscript in $D_{max}$ and $sZDR_{max}$. Fixed

- page 12 / line 240: mueller -> Mueller. We have fixed all occurrences of mueller to Mueller and subsequently also amplitude to Amplitude.

- page 21 / line 451: change Dmax into $D_\text{max}$. We have fixed all occurrences in the text.

- caption of Fig. 10: subscript "max" in sZDRmax (with upright font). Fixed

- page 17, line 386: "e.g." instead of "i.e."?. Fixed

- page 18, line 390: "describes" verb (used twice) seems misleading, I'd suggest, e.g., "is linked with" (as it is rather the mass flux proportionality to "S-1" that describes the diffusion, while the Fk+Fd term originates from an approximation simplifying the coupled mass/heat diffusion system into a single mass diffusion equation). Changed to "is linked with"

- page 19: 13 occurrences of non-math short hyphen instead of a minus sign in temperature values
- page 21: ditto - 4 occurrences
- page 22: ditto - 5 occurrences. Fixed all occurrences in the text.
- page 40: ditto - 2 occurrences (in title of Bailey and Hallett). Fixed in the reference

- page 23 / line 495: suggest adding the unit (incl. degree sign) to the 0.1 value as well. Added the unit.

- reference Doviak and Zrnic 1993 reference has an invalid DOI URL (https://doi.org/https://...) and lacks capitalization in publisher name. Fixed
- reference Mitchell et al. 1996 has an invalid DOI URL (https://doi.org/https://...) and a lack capitalization in journal name. Fixed
- reference Ori and Kneifel 2018 has an invalid DOI URL (https://doi.org/https://...). Fixed
- references Schrom and Kumjian 2018, Tyynelä et al. 2009 is missing a DOI. Fixed
- reference Mätzler 2006 is missing book series name to make the "vol." understandable ("IET ELECTROMAGNETIC WAVES SERIES"), also "London" could be added as "address" field. Added the book series name
- reference Mishchenko et al. 2002: fix capitalization in publisher name. Fixed
- reference Purcell and Pennypacker 1973 is missing a DOI. Fixed
- reference von Terzi et al. 2025 needs a more elaborate information than just "level1-data". Added more information
- reference Yurkin et al. 2006: journal name is abbreviated, while elsewhere not. Fixed
- references Zubko et al. 2010, Westbrook 2004b: journal name capitalization. Fixed
- reference Mason et al. 2019: final revised paper should be cited instead of the preprint. Fixed